# Ambient Diffusion Posterior Sampling: Solving Inverse Problems with Diffusion Models trained on Corrupted Data

**Asad Aali**[*]
Stanford University
UT Austin

**Giannis Daras**[*]
MIT CSAIL
IFML

**Brett Levac**
UT Austin
IFML

**Sidharth Kumar**
UT Austin
IFML

**Alexandros G. Dimakis**
UC Berkeley
Bespoke Labs

**Jonathan I. Tamir**
UT Austin
IFML

## Abstract

We provide a framework for solving inverse problems with diffusion models learned from linearly corrupted data. Firstly, we extend the Ambient Diffusion framework to enable training directly from measurements corrupted in the Fourier domain. Subsequently, we train diffusion models for MRI with access only to Fourier sub-sampled multi-coil measurements at acceleration factors R= $2, 4, 6, 8$. Secondly, we propose *Ambient Diffusion Posterior Sampling* (A-DPS), a reconstruction algorithm that leverages generative models pre-trained on one type of corruption (e.g. image inpainting) to perform posterior sampling on measurements from a different forward process (e.g. image blurring). For MRI reconstruction in high acceleration regimes, we observe that A-DPS models trained on subsampled data are better suited to solving inverse problems than models trained on fully sampled data. We also test the efficacy of A-DPS on natural image datasets (CelebA, FFHQ, and AFHQ) and show that A-DPS can sometimes outperform models trained on clean data for several image restoration tasks in both speed and performance.

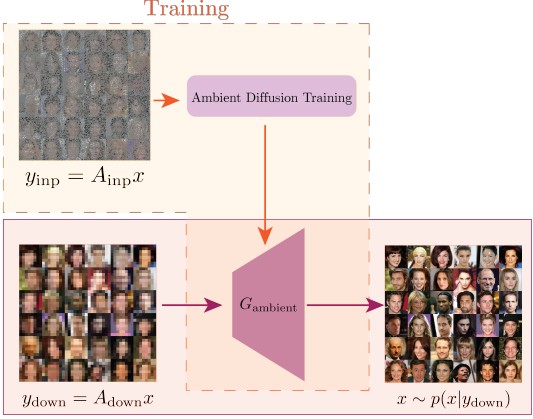

Figure 1: Ambient Diffusion Posterior Sampling (Ambient DPS). During training, we only have access to linearly corrupted data from a forward operator $A_{\text{train}}$. We use the Ambient Diffusion framework to learn a generative model, $G_{\text{ambient}}$, for the uncorrupted distribution, $p(\boldsymbol{x}_0)$. At inference time, we sample from the posterior distribution $p(\boldsymbol{x}_0|\boldsymbol{y}_{A_{\text{inf}}})$, for measurements $\boldsymbol{y}_{\text{inf}}$ coming from a different forward operator, $A_{\text{inf}}$.

## 1 Introduction

For some applications, it is expensive or impossible to acquire fully observed data (Collaboration et al., 2019; Gao et al., 2023; Yaman et al., 2020) but possible to acquire partially observed samples.

---

[*]Equal contribution.

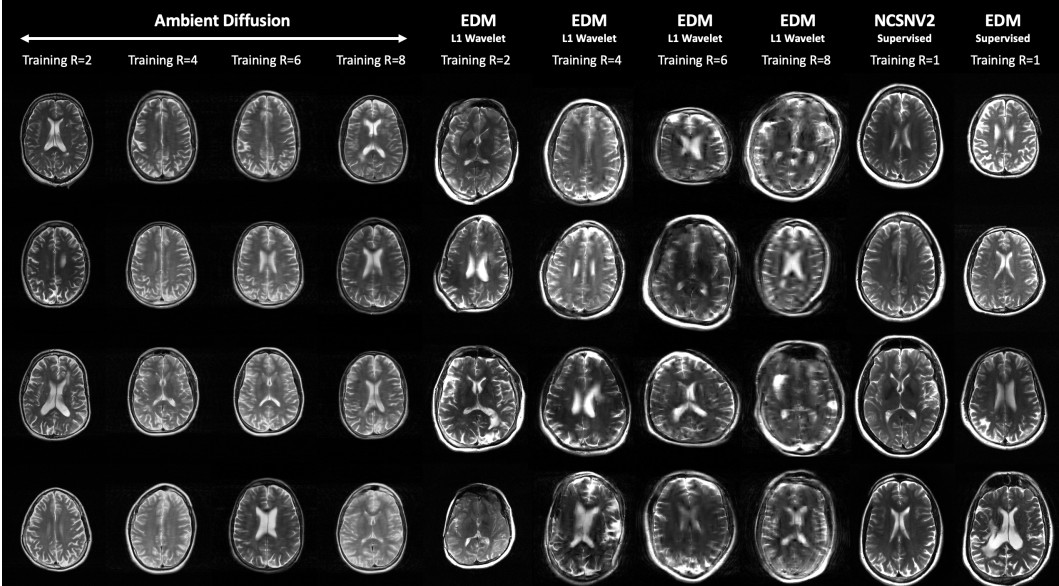

Figure 2: Prior samples from diffusion models trained on MRI scans. Columns $1 - 4$: Diffusion models trained on subsampled MRI scans at acceleration factors $R = 2, 4, 6, 8$, using the Ambient Diffusion framework extended for Fourier subsampled training. Columns $5 - 8$: EDM models trained with L1-wavelet reconstructions of subsampled scans at $R = 2, 4, 6, 8$. Column 9: NCSNV2 trained with fully sampled scans. Column 10: EDM trained with fully sampled scans. We observe that Ambient Diffusion models consistently produce high-quality and realistic MRI scans even in high acceleration regimes.

Furthermore, it may be desirable to train generative models with corrupted data since that reduces memorization of the training set (Daras et al., 2023b; Carlini et al., 2023; Somepalli et al., 2022). Prior works have shown how to train Generative Adversarial Networks (GANs) (Bora et al., 2018; Cole et al., 2021), flow models (Kelkar et al., 2023) and restoration models (Lehtinen et al., 2018; Krull et al., 2019; Yaman et al., 2020; Millard & Chiew, 2023) with corrupted data. More recently, there has been a shift towards training *diffusion generative models* given corrupted data (Daras et al., 2024; 2023a;b; Aali et al., 2023; Kawar et al., 2023; Cui et al., 2022; Kim & Ye, 2021). What remains unexplored is how generative models trained on a certain type of corruption (e.g. inpainted data) can be used to solve inverse problems with a different forward process (e.g. blurring).

We propose the first framework to solve inverse problems with Ambient Diffusion models (Daras et al., 2023b). These models are trained using only access to linear measurements and they estimate the *ambient score*, i.e. how to best reconstruct given an input with a corrupted linear forward operator *corrupted noisy input*. We show how to use these models for solving linear inverse problems outside their training distribution. Our experiments on datasets of natural images and multi-coil MRI show something surprising: Ambient Models can outperform (in the high corruption regime) models trained on clean data while being substantially faster. Our algorithm extends Diffusion Posterior Sampling (Chung et al., 2023) to Ambient Diffusion models. **Our contributions:**[†]

- We propose *Ambient Diffusion Posterior Sampling* (Ambient DPS), an algorithm that uses diffusion models trained on linearly corrupted data as priors for solving inverse problems with arbitrary linear measurement models.
- We extend the Ambient Diffusion framework to train models using Fourier subsampled measurements. Specifically, we train on subsampled multi-coil MRI scans at various retrospective acceleration factors (R=$2, 4, 6, 8$); we observe that models trained on subsampled data are better priors for solving inverse problems in the high acceleration regime.
- We use pre-trained Ambient Diffusion models to solve inverse problems (compressed sensing, super-resolution) on natural image datasets (CelebA, FFHQ, AFHQ) and show that they can even outperform models trained on clean data in the high corruption regime.

---

[†]We open-source our code and Ambient Diffusion models: github.com/utcsilab/ambient-diffusion-mri.

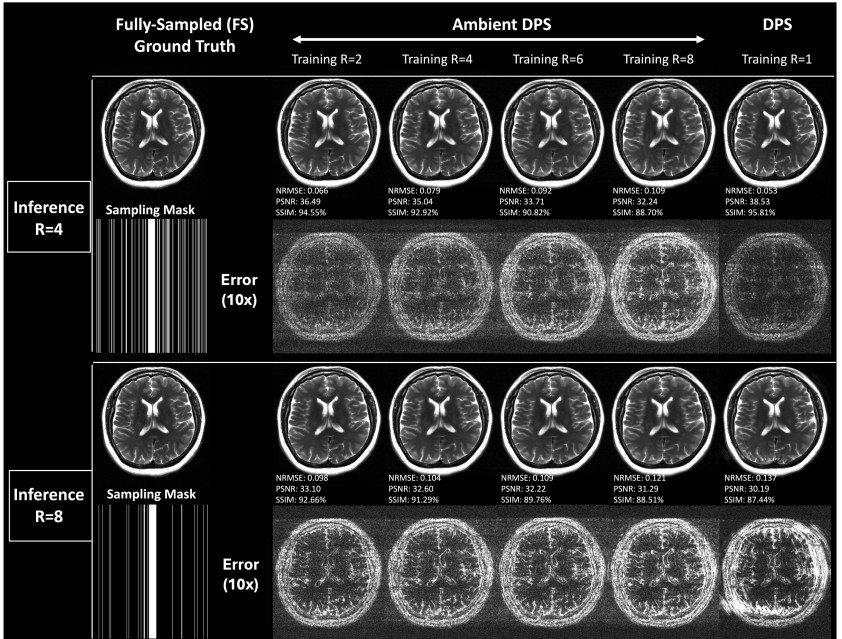

Figure 3: Posterior sampling reconstructions for MRI scans using models trained on Fourier subsampled data at various acceleration factors (Ambient DPS, columns $2-5$) and a model trained on clean data (FS-DPS (Chung et al., 2023), column 6). Rows 1 and 3 show reconstructions at $R = 4$ and $R = 8$, respectively, while rows 2 and 4 display the difference to the ground truth on a $10\times$ scale. At high inference acceleration (R=8), *Ambient DPS*, outperforms FS-DPS, despite that the underlying models were trained solely on corrupted data.

## 2 METHOD

### 2.1 BACKGROUND AND NOTATION

**Diffusion Posterior Sampling (DPS).** Diffusion models are typically trained to reconstruct a clean image $\boldsymbol{x}_0 \sim p_0(\boldsymbol{x}_0)$ from a noisy observation $\boldsymbol{x}_t = \boldsymbol{x}_0 + \sigma_t\boldsymbol{\eta}, \quad \eta \sim \mathcal{N}(\mathbf{0}, I)$. Despite the simplicity of the training objective, diffusion models can approximately sample from $p(\boldsymbol{x})$ by running a discretized version of the Stochastic Differential Equation (Song et al., 2020; Anderson, 1982):

$$\mathrm{d}\boldsymbol{x} = -2\dot{\sigma}_t(\mathbb{E}[\boldsymbol{x}_0|\boldsymbol{x}_t] - \boldsymbol{x}_t)\mathrm{d}t + g(t)\mathrm{d}\boldsymbol{w}, \tag{2.1}$$

where $\boldsymbol{w}$ is the standard Wiener process and $\mathbb{E}[\boldsymbol{x}_0|\boldsymbol{x}_t]$ is estimated by the network. Given measurement $\boldsymbol{y}_{\mathrm{inf}} = A_{\mathrm{inf}}\boldsymbol{x}_0$, one can sample from the posterior $p(\boldsymbol{x}_0|\boldsymbol{y}_{\mathrm{inf}})$ by running the process:

$$\mathrm{d}\boldsymbol{x} = -2\dot{\sigma}_t\sigma_t \left( \frac{\mathbb{E}[\boldsymbol{x}_0|\boldsymbol{x}_t] - \boldsymbol{x}_t}{\sigma_t} + \underbrace{\nabla \log p(\boldsymbol{y}_{\mathrm{inf}}|\boldsymbol{x}_t)}_{\text{likelihood term}} \right) \mathrm{d}t + g(t)\mathrm{d}\boldsymbol{w}. \tag{2.2}$$

For most forward operators it is intractable to write the likelihood in closed form. Hence, several approximations have been proposed to use diffusion models for inverse problems (Chung et al., 2023; Kawar et al.; Jalal et al., 2021; Song et al., 2021; Chung et al., 2022; Feng et al., 2023; Graikos et al., 2022). One of the simplest and most effective approximations is Diffusion Posterior Sampling (DPS) (Chung et al., 2023). DPS estimates $\boldsymbol{x}_0$ using $\boldsymbol{x}_t$ and uses the conditional likelihood $p(\boldsymbol{y}_{\mathrm{inf}}|\hat{\boldsymbol{x}}_0)$ instead of the intractable term, i.e. DPS approximates $p(\boldsymbol{y}_{\mathrm{inf}}|\boldsymbol{x}_t)$ with $p(\boldsymbol{y}_{\mathrm{inf}}|\boldsymbol{x}_0 = \mathbb{E}[\boldsymbol{x}_0|\boldsymbol{x}_t])$, where $\gamma_t$ is a tunable guidance parameter. The update rule becomes:

$$\mathrm{d}\boldsymbol{x} = -2\dot{\sigma}_t\sigma_t \left( \frac{\mathbb{E}[\boldsymbol{x}_0|\boldsymbol{x}_t] - \boldsymbol{x}_t}{\sigma_t} + \gamma_t\nabla_{\boldsymbol{x}_t} \log p(\boldsymbol{y}_{\mathrm{inf}}|\boldsymbol{x}_0 = \mathbb{E}[\boldsymbol{x}_0|\boldsymbol{x}_t]) \right) \mathrm{d}t + g(t)\mathrm{d}\boldsymbol{w}, \tag{2.3}$$

**Ambient Diffusion.** In some settings we do not have a large training set of clean data but we have access to a large set of lossy measurements that we would like to leverage to train a diffusion model for the clean distribution.

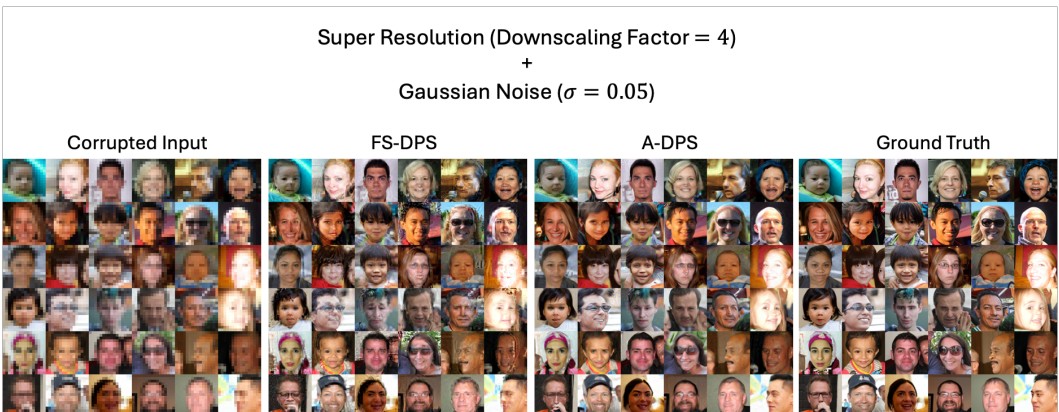

Figure 4: Posterior sampling reconstructions for FFHQ, showing: (1) Corrupted Input: Examples from FFHQ dataset corrupted using Resolution Downscaling ($4\times$) and Gaussian Noise ($\sigma = 0.05$), (2) FS-DPS: Reconstruction using Diffusion Posterior Sampling (DPS), with a diffusion model trained with clean fully-sampled data ($p = 0.0$), (3) A-DPS: Reconstruction using Ambient-DPS, with a diffusion model trained on randomly inpainted data with erasure probability ($p = 0.6$), (4) Ground Truth: Original uncorrupted examples from FFHQ dataset. We observe that *Ambient DPS*, provides better reconstructions (qualitatively) even though the underlying models were trained on corrupted data ($p = 0.6$).

The authors of Daras et al. (2023b) consider the setting of having access to linearly corrupted data $\{\boldsymbol{y}_0 = A_{\text{train}}x_0, A_{\text{train}}\}$, where the distribution of $A_{\text{train}}$, denoted as $p(A_{\text{train}})$, is assumed to be known. The ultimate goal is to learn the best restoration model given a linearly corrupted noisy input $\boldsymbol{y}_{t,\text{train}} = A_{\text{train}}(\boldsymbol{x}_0 + \sigma_t\boldsymbol{\eta})$, at all noise levels $t$. The authors of Daras et al. (2023b) form further corrupted iterates $\tilde{\boldsymbol{y}}_{t,\text{train}} = \tilde{A}_{\text{train}}(\boldsymbol{x}_0 + \sigma_t\boldsymbol{\eta})$ using a matrix $\tilde{A}_{\text{train}}$ (that is a perturbation of the given matrix $A_{\text{train}}$) and train with the following objective:

$$J(\theta) = \mathbb{E}_{\boldsymbol{x}_0, \boldsymbol{x}_t, A_{\text{train}}, \tilde{A}_{\text{train}}} \left[ \left\| A_{\text{train}}\boldsymbol{h}_\theta(\tilde{\boldsymbol{y}}_{0,\text{train}}, \tilde{A}_{\text{train}}) - \boldsymbol{y}_{t,\text{train}} \right\| \right], \tag{2.4}$$

that provably learns $\mathbb{E}[\boldsymbol{x}_0 | \tilde{A}_{\text{train}}, \tilde{\boldsymbol{y}}_{t,\text{train}}]$ as long as the matrix $\mathbb{E}[A_{\text{train}}^T A_{\text{train}} | \tilde{A}_{\text{train}}]$ is full-rank. In certain cases, it is possible to introduce *minimal* additional corruption and satisfy this condition. For example, if $A_{\text{train}}$ is a random inpainting matrix (i.e. $A_{ij} \sim \text{Be}(1 - p)$), then $\tilde{A}_{\text{train}}$ can be formed by taking $A_{\text{train}}$ and erasing additional pixels with any non-zero probability $\delta > 0$.

**Multi-coil Magnetic Resonance Imaging.** MRI is a prototypical use case for a framework that can learn generative models from linearly corrupted data, as in many cases it is not feasible to collect a large training set of fully sampled data (Tibrewala et al., 2023; Desai et al., 2022; Tariq et al.). In settings such as 3D+time dynamic contrast-enhanced MRI (Zhang et al., 2015) it is impossible to collect fully sampled data due to the time-varying dynamics of the contrast agent (Zhang et al.).

In the multi-coil MRI setting, the acquisition involves collecting measurements of an image directly in the spatial frequency, known as $k$-space, from a set of spatially localized coils. Mathematically, there are $N_c$ coils, each of which gives measurements:

$$\boldsymbol{z}_{\boldsymbol{x},i} = P\mathcal{F}S_i\boldsymbol{x} + \boldsymbol{w}_i, \quad i \in [N_c], \tag{2.5}$$

where $\boldsymbol{x}$ is the (complex-valued) image of interest, $S_i$ represents the coil-sensitivity profile of the $i^{th}$ coil, $\mathcal{F}$ is the Fourier transform, $P$ represents the Fourier subsampling operator and $\boldsymbol{w}_i$ is complex-valued Gaussian i.i.d noise. We assume the noiseless case and we point the reader to Daras et al. (2023a); Aali et al. (2023); Kawar et al. (2023) for approaches that handle the noisy case.

For the discrete approximation of the continuous signal as an image $\boldsymbol{x} \in C^n$, the composition of $P, \mathcal{F}, S_i$ can be written as a matrix $A_i \in C^{m \times n}$, where the number of measurements, $m$, depends on the subsampling operator $P$. It is common to denote with $R$ the ratio $\frac{n}{m}$, which is known as the acceleration factor. At inference time (i.e., for a new patient), we typically want to acquire data with a high acceleration factor because this reduces scan time and patient discomfort.

Due to the time required to collect $k$-space measurements, it is often not possible to acquire fully sampled $k$-space. Hence, training cannot rely on a fully sampled image to guide reconstruction

quality as is done in the fully supervised setting (Aggarwal et al., 2019; Hammernik et al., 2018). If two independent measurements are available for the same underlying signal, restoration models can be trained without having access to uncorrupted data (Lehtinen et al., 2018; Gan et al., 2023). If only one measurement is available (as in our setting), some end-to-end techniques use a loss on the measurement domain by partitioning the training measurements for each sample into (1) measurements for reconstruction, and (2) measurements for applying the loss function (Yaman et al., 2020). Other approaches leverage structure in the MRI acquisition to learn from limited-resolution data (Wang et al., 2023). More recently, works have begun leveraging signal set properties such as group invariance to assist in learning from subsampled measurements for a variety of inverse problems (Tachella et al., 2022b; Scanvic et al., 2023; Chen et al., 2022a; Tachella et al., 2022a). The idea of additional corruption (as we do in our work), is more related to the Noisier2Noise framework (Moran et al., 2020) which has been initially developed for denoising. Millard & Chiew (2023) extended this idea to the MRI setting to learn models for MRI restoration without access to clean data. All these approaches learn a restoration model without access to reference data. However, as they are inherently end-to-end methods, their performance on out-of-distribution tasks (e.g., due to different acquisition trajectories) is known to degrade (Jalal et al., 2021; Zach et al., 2022).

## 2.2 Ambient Diffusion for MRI

The MRI acquisition process results in linearly corrupted measurements and thus it should be possible to use the Ambient Diffusion framework to learn from corrupted observations. Yet, we identified three important changes that differ from the setting studied by Daras et al. (2023b): i) the inpainting happens in the Fourier Domain, ii) the inpainting has structure, i.e. whole vertical lines in the spatial Fourier are either observed or not observed (instead of masking random pixels) and, iii) the image is measured from an array of spatially varying receive coils. In what follows, we show how to account for these factors and apply the Ambient Diffusion framework to MRI data.

First, as in Ambient Diffusion, we will corrupt further the given measurements, but in our case, the additional corruption will happen in the Fourier space. We create further corrupted measurements for each coil by decreasing the Fourier subsampling ratio, i.e. we create iterates:

$$\tilde{z}_{\boldsymbol{x},i} = \tilde{P}\mathcal{F}S_i\boldsymbol{x}. \tag{2.6}$$

This is possible to do because we can just subsample the available data $z_{\boldsymbol{x},i}$. To further corrupt the given measurements, we keep the inpainting structure that there is in the measurements, i.e. we delete (at random) 2-D lines in Fourier space (instead of pixel masking as in Daras et al. (2023b)). Due to the multiplicity of coils, we combine their measurements to form a crude estimator of $\boldsymbol{x}$ by taking the adjoint of $\tilde{A}_{\text{train}}$: $\tilde{\boldsymbol{y}}_{\text{train}} = \sum_i S_i^H \mathcal{F}^{-1}(\tilde{\boldsymbol{z}}_{\boldsymbol{x},i})$. Notice that for a discrete signal $\boldsymbol{x} \in \mathbb{C}^n$, all these operations are linear and hence $\tilde{\boldsymbol{y}}$ can be written as:

$$\tilde{\boldsymbol{y}}_{\text{train}} = \underbrace{\left( \sum_i S_i^H \mathcal{F}^{-1}\tilde{P}\mathcal{F}S_i \right)}_{\tilde{A}_{\text{train}}} \boldsymbol{x}. \tag{2.7}$$

As in Ambient Diffusion, we will train the network to predict the corresponding signal before the additional corruption, i.e. our target will be:

$$\boldsymbol{y}_{\text{train}} = \underbrace{\left( \sum_i S_i^H \mathcal{F}^{-1}P\mathcal{F}S_i \right)}_{A_{\text{train}}} \boldsymbol{x}. \tag{2.8}$$

We are now ready to state our main Theorem.

**Theorem 2.1** ((Informal)). *Let $\boldsymbol{y}_{t,\text{train}}, \tilde{\boldsymbol{y}}_{t,\text{train}}$ represent the noisy versions of $\boldsymbol{y}_{\text{train}}, \tilde{\boldsymbol{y}}_{\text{train}}$ respectively, i.e.:*

$$\begin{cases} \boldsymbol{y}_{t,\text{train}} = A_{\text{train}} \left( \boldsymbol{x} + \sigma_t \boldsymbol{\eta} \right) \\ \tilde{\boldsymbol{y}}_{t,\text{train}} = \tilde{A}_{\text{train}} \left( \boldsymbol{x} + \sigma_t \boldsymbol{\eta} \right) \end{cases}. \tag{2.9}$$

*Then, the minimizer of the objective:*

$$J(\theta) = \mathbb{E}_{\boldsymbol{y}_{0,\text{train}}, \tilde{\boldsymbol{y}}_{t,\text{train}}, A_{\text{train}}, \tilde{P}} \left[ \left| \left| A_{\text{train}} \boldsymbol{h}_\theta(\tilde{\boldsymbol{y}}_{t,\text{train}}, \tilde{P}) - \boldsymbol{y}_{0,\text{train}} \right| \right|^2 \right], \tag{2.10}$$

*is: $\boldsymbol{h}_\theta(\tilde{\boldsymbol{y}}_{t,\text{train}}, \tilde{P}) = \mathbb{E}[\boldsymbol{x}_0 | \tilde{\boldsymbol{y}}_{t,\text{train}}, \tilde{P}]$.*

**Proof overview.** The formal proof of this Theorem is given in the Appendix. We provide a sketch of the proof here. The techniques are based on Theorem 4.2 of Ambient Diffusion (Daras et al., 2023b). In Theorem 4.2, the condition that needs to be satisfied is that $\mathbb{E}[A_{\text{train}}^T A_{\text{train}} | \tilde{A}_{\text{train}}]$ is full-rank. Our objective is slightly different because the network doesn't see the whole forward operator $\tilde{A}_{\text{train}}$, but only part of it, i.e. the matrix $\tilde{P}$. By following the steps of the Ambient Diffusion proof, we arrive at the necessary condition for our case, which is to prove that $\mathbb{E}[A_{\text{train}} | \tilde{P}]$ is full-rank.

We start the argument by noting that $\mathbb{E}[P | \tilde{P}]$ is full-rank. Intuitively, this is because there is a non-zero probability of observing any Fourier coefficient and hence any deleted Fourier co-efficient could be due to the extra corruption introduced by $\tilde{P}$. Since $\mathcal{F}$ is an invertible matrix, then also $\mathbb{E}[\mathcal{F}^{-1} P \mathcal{F} | \tilde{P}]$ is full-rank. Finally, we use the properties of the sensitivity masks to show that $\mathbb{E}\left[\sum_i S_i^H \mathcal{F}^{-1} P \mathcal{F} S_i | \tilde{P}\right]$ is also full-rank and this completes the proof.

## 2.3 AMBIENT DIFFUSION POSTERIOR SAMPLING

DPS requires access to $\mathbb{E}[\boldsymbol{x}_0 | \boldsymbol{x}_t]$ to approximately sample from $p(\boldsymbol{x}_0 | \boldsymbol{y}_{\text{inf}})$. Since Ambient Diffusion models can only work with corrupted inputs, we propose the following update rule instead:

$$\mathrm{d}\boldsymbol{x} = -2\dot{\sigma}_t \sigma_t \left( \underbrace{\frac{\mathbb{E}[\boldsymbol{x}_0 | \tilde{\boldsymbol{y}}_{t,\text{train}}, \tilde{A}_{\text{train}}] - \boldsymbol{x}_t}{\sigma_t}}_{\text{Ambient Score}} + \gamma_t \nabla_{\boldsymbol{x}_t} \log p(\boldsymbol{y}_{\text{inf}} | \boldsymbol{x}_0 = \mathbb{E}[\boldsymbol{x}_0 | \tilde{\boldsymbol{y}}_{t,\text{train}}, \tilde{A}_{\text{train}}]) \right) \mathrm{d}t + g(t)\mathrm{d}\boldsymbol{w},$$

(2.11)

for a fixed $\tilde{A}_{\text{train}} \sim p(\tilde{A}_{\text{train}})$ [*]. Comparing this to the DPS update rule (E.q. 2.3), all the $\mathbb{E}[\boldsymbol{x}_0 | \boldsymbol{x}_t]$ terms have been replaced with their ambient counterparts, i.e. with $\mathbb{E}[\boldsymbol{x}_0 | \tilde{\boldsymbol{y}}_{t,\text{train}}, \tilde{A}_{\text{train}}]$. The latter is approximated by a neural network that is trained with Ambient Diffusion.

We term our approximate sampling algorithm for solving inverse problems with diffusion models learned from corrupted data **Ambient DPS (A-DPS)**. We remark that similar to DPS, the proposed algorithm is an approximation to sampling from the true posterior distribution. Similarly to Theorem 1 of the DPS paper, for a given measurement $\boldsymbol{y}_{\text{inf}}$ at noise level $\sigma_{\boldsymbol{y}}$ and a given matrix $A_{\text{inf}}$, one can upper-bound the Jensen gap of A-DPS as follows:

$$\mathcal{J} \le \frac{n}{\sqrt{2\pi\sigma_{\boldsymbol{y}}^2}} \exp(-1/2\sigma_{\boldsymbol{y}}^2) ||A_{\text{inf}}|| \int_{\boldsymbol{x}_0} ||\boldsymbol{x}_0 - \hat{\boldsymbol{x}}_0|| p(\boldsymbol{x}_0 | \boldsymbol{x}_t) \mathrm{d}\boldsymbol{x}_0,$$

(2.12)

where $\hat{\boldsymbol{x}}_0 = \mathbb{E}[\boldsymbol{x}_0 | \tilde{A}_{\text{train}} \boldsymbol{x}_t, \tilde{A}_{\text{train}}]$, for a fixed $\tilde{A}_{\text{train}}$ sampled from $p(\tilde{A}_{\text{train}})$. We omit the proof of this proposition since it follows the exact same steps as the proof of Theorem 1 in the DPS paper.

## 2.4 AMBIENT DIFFUSION FOR IN-DOMAIN RECONSTRUCTIONS.

Ambient DPS can be used to solve *any* inverse problem for which the forward operator is known, using a diffusion model trained on linearly corrupted data of some form. It is important to underline though that if the inverse problem that we want to solve at inference time has the same forward operator as the one that was used for the training measurements, then we can use the Ambient Diffusion as a supervised restoration model. This is because Ambient Diffusion models are trained to estimate $\mathbb{E}[\boldsymbol{x}_0 | \tilde{\boldsymbol{y}}_{\text{train}}, \tilde{A}_{\text{train}}]$, and hence if $A_{\text{inf}}$ comes from the same distribution as $A_{\text{train}}$, then, the one-step prediction of the model, Ambient One Step (A-OS), is the Mean Squared Error (MSE) minimizer. Similarly, if we are interested in unconditional generation, we can simply run Ambient DPS without the likelihood term.

---

[*]The Ambient MRI models take as input $\tilde{P}$ instead of $\tilde{A}$, as in Equation 2.10.

Table 1: Unconditional and Conditional Sampling FID for models trained on MRI data at different acceleration factors.

| Training Data | Training Method | Unconditional FID ↓ |
|---|---|---|
| $R = 1$ | EDM | **10.41** |
|  | NCSNV2 | 13.20 |
| $R = 2$ | L1-EDM | **18.55** |
|  | Ambient Diffusion | 30.34 |
| $R = 4$ | L1-EDM | **27.64** |
|  | Ambient Diffusion | 32.31 |
| $R = 6$ | L1-EDM | 51.43 |
|  | Ambient Diffusion | **31.50** |
| $R = 8$ | L1-EDM | 102.98 |
|  | Ambient Diffusion | **48.15** |

| Training Data | Inference Data | Reconstruction Method | Conditional FID ↓ |
|---|---|---|---|
| $R = 1$ | $R = 8$ | FS-DPS | 4.16 |
| $R = 2$ | | | **0.71** |
| $R = 4$ | $R = 8$ | A-DPS (*our method*) | 0.87 |
| $R = 6$ | | | 1.29 |
| $R = 8$ | | | 2.00 |

## 3 EXPERIMENTS

### 3.1 AMBIENT MRI DIFFUSION MODELS

In this section, we extend Ambient Diffusion to the multi-coil Fourier subsampled MRI setting (as detailed in Section 2.2) and we train our own MRI models from scratch. We give details about our dataset preparation in Section B in the Appendix.

**Sampling masks.** We retrospectively subsample the $k$-space training data by applying randomly subsampled masks that correspond to acceleration factors $R \in \{2, 4, 6, 8\}$. The sampling masks include fully sampled vertical (readout) lines corresponding to the observed Fourier coefficients (phase encodes). We always sample the central 20 lines for autocalibration. To form further corrupted measurements, we randomly remove additional lines such that we create measurements at acceleration factor $R + 1$. For example, we take data at $R = 2$, we create further corrupted measurements at $R = 3$ and we train the model to predict the clean image by measuring its error with the available data at $R = 2$ given its prediction for the $R = 3$ corrupted input.

**Comparison Models.** We train one model for each acceleration factor $R \in \{1, 2, 4, 6, 8\}$. The $R = 1$ model is trained on clean data (no extra corruption) based on the EDM approach (Karras et al., 2022). The $R > 1$ models are trained with our modified Ambient Diffusion framework. While we focus on EDM for fully sampled training, we also train an NCSNV2 model for $R = 1$ following the approach in Jalal et al. (2021). As baselines, we also train EDM models after L1-Wavelet compressed sensing reconstruction (L1-EDM) of the training set at each acceleration factor (Lustig et al., 2007).

**Unconditional generation evaluation.** We evaluate the unconditional generation performance of each model and show prior samples for Ambient Diffusion and EDM models at $R = 1$ in Figure 2. Similar to the results in Daras et al. (2023b), we observe that the Ambient Diffusion samples are qualitatively similar to EDM models at low acceleration, and become slightly blurry at higher accelerations. Notably, the samples generated by Ambient Diffusion at $R = 8$ have no residual aliasing artifacts, in contrast to the samples generated from L1-Wavelet reconstructions.

To quantify unconditional sample quality, we follow the approach in Bendel et al. (2022) and calculate FID scores from 100 samples using a pre-trained VGG network. Table 1, shows FID scores for the diffusion models. While there is an increase in FID for the Ambient models, we see that those trained at higher acceleration factors outperform L1-EDM models trained on L1-Wavelet reconstructions.

| Training Method | Training Data | Reconstruction Method | R = 2 | | | R = 4 | | | R = 6 | | | R = 8 | | |
|---|---|---|---|---|---|---|---|---|---|---|---|---|---|---|
| | | | NRMSE | SSIM | PSNR | NRMSE | SSIM | PSNR | NRMSE | SSIM | PSNR | NRMSE | SSIM | PSNR |
| Supervised | $R=1$ | FS-DPS | **0.034** | **97.83** | **43.70** | **0.065** | **95.20** | **38.17** | 0.121 | 90.60 | 32.99 | 0.179 | 85.21 | 29.46 |
| | | FS-ALD | 0.059 | 94.55 | 40.17 | 0.092 | 90.71 | 36.13 | 0.122 | 88.53 | 33.62 | 0.157 | 86.11 | 31.22 |
| | | L1-CS | 0.092 | 91.10 | 36.13 | 0.143 | 82.08 | 31.58 | 0.217 | 77.07 | 27.74 | 0.259 | 75.04 | 26.09 |
| | | MoDL ($R=2$) | 0.036 | 97.56 | 43.03 | 0.076 | 94.34 | 36.65 | 0.132 | 87.56 | 31.84 | 0.182 | 81.13 | 29.01 |
| | | MoDL ($R=4$) | 0.041 | 96.67 | 41.95 | 0.066 | 95.19 | 37.85 | 0.095 | 93.09 | 34.62 | 0.124 | 90.81 | 32.31 |
| | | MoDL ($R=6$) | 0.045 | 95.37 | 41.04 | 0.067 | 94.83 | 37.65 | **0.090** | **93.41** | **35.04** | **0.111** | **91.87** | **33.21** |
| | | MoDL ($R=8$) | 0.050 | 94.13 | 40.24 | 0.079 | 94.30 | 36.20 | 0.096 | 90.09 | 34.46 | 0.113 | 91.08 | 33.08 |
| Self-Supervised | $R=2$ | A-DPS (*our method*) | 0.058 | 94.88 | 38.94 | **0.074** | **93.99** | **36.82** | **0.093** | 92.90 | 34.84 | **0.112** | 91.63 | 33.19 |
| | | A-ALD (*our method*) | 0.058 | 94.84 | 38.89 | **0.074** | **93.99** | **36.82** | **0.093** | 92.83 | 34.79 | 0.114 | 91.52 | 33.07 |
| | | A-OS | 0.076 | 94.93 | 36.42 | 0.111 | 91.81 | 33.19 | 0.159 | 88.04 | 30.04 | 0.196 | 84.85 | 28.26 |
| | | L1-DPS | **0.036** | **97.54** | **43.20** | 0.088 | 93.35 | 35.64 | 0.151 | 87.57 | 30.94 | 0.205 | 81.84 | 28.09 |
| | | SSDU | 0.057 | 94.80 | 39.25 | 0.134 | 88.55 | 31.76 | 0.199 | 82.09 | 28.23 | 0.247 | 77.53 | 26.35 |
| | | 1D-Partitioned SSDU | 0.076 | 94.94 | 36.38 | 0.083 | 90.47 | 35.61 | 0.107 | 86.14 | 33.45 | 0.128 | 82.82 | 31.9 |
| | $R=4$ | A-DPS (*our method*) | 0.072 | 93.31 | 37.0 | **0.084** | 92.82 | 35.67 | **0.099** | 91.98 | 34.24 | **0.113** | 91.06 | 33.05 |
| | | A-ALD (*our method*) | 0.071 | 93.32 | 37.03 | **0.084** | 92.77 | 35.64 | **0.099** | **91.98** | **34.24** | **0.113** | **91.13** | **33.08** |
| | | A-OS | 0.108 | 91.25 | 31.95 | 0.105 | 91.83 | 32.81 | 0.136 | 89.90 | 31.60 | 0.162 | 87.74 | 30.43 |
| | | L1-DPS | **0.040** | **96.59** | **42.45** | 0.112 | 89.98 | 33.58 | 0.167 | 84.68 | 29.98 | 0.211 | 80.26 | 27.77 |
| | | SSDU | 0.072 | 87.84 | 37.20 | 0.088 | **92.82** | 35.36 | 0.131 | 89.74 | 31.81 | 0.167 | 87.01 | 29.72 |
| | | 1D-Partitioned SSDU | 0.099 | 93.45 | 34.05 | 0.103 | 89.79 | 33.71 | 0.114 | 85.96 | 32.84 | 0.127 | 82.96 | 31.91 |
| | $R=6$ | A-DPS (*our method*) | 0.084 | 91.85 | 35.63 | **0.094** | 91.47 | **34.62** | 0.106 | **90.87** | **33.60** | 0.118 | **90.20** | **32.69** |
| | | A-ALD (*our method*) | 0.084 | 91.85 | 35.65 | 0.095 | 91.42 | 34.56 | 0.108 | 90.74 | 33.48 | 0.119 | 90.12 | 32.60 |
| | | A-OS | 0.127 | 89.83 | 31.95 | 0.116 | 90.25 | 32.81 | 0.134 | 89.45 | 31.60 | 0.153 | 88.16 | 30.43 |
| | | L1-DPS | **0.043** | **95.94** | **41.73** | 0.112 | 88.67 | 32.72 | 0.175 | 83.68 | 29.49 | 0.215 | 79.75 | 27.59 |
| | | SSDU | 0.075 | 89.72 | 36.84 | 0.105 | **91.77** | 33.82 | 0.138 | 89.42 | 31.34 | 0.166 | 87.29 | 29.76 |
| | | 1D-Partitioned SSDU | 0.081 | 94.44 | 35.81 | 0.103 | 89.60 | 33.73 | 0.121 | 85.52 | 32.37 | 0.134 | 82.47 | 31.48 |
| | $R=8$ | A-DPS (*our method*) | 0.099 | 90.51 | 34.15 | **0.107** | **90.47** | **33.46** | **0.117** | **89.94** | **32.71** | 0.126 | **89.50** | **32.10** |
| | | A-ALD (*our method*) | 0.100 | 90.47 | 34.11 | **0.107** | 90.44 | **33.46** | 0.118 | 89.81 | 32.63 | 0.127 | 89.29 | 32.00 |
| | | A-OS | 0.135 | 86.46 | 31.46 | 0.136 | 87.08 | 31.46 | 0.152 | 86.95 | 30.53 | 0.164 | 86.54 | 29.86 |
| | | L1-DPS | **0.044** | **95.94** | **41.53** | 0.126 | 88.47 | 32.41 | 0.183 | 82.93 | 29.07 | 0.223 | 78.94 | 27.24 |
| | | SSDU | 0.118 | 77.19 | 32.95 | 0.122 | 88.15 | 32.42 | 0.151 | 87.93 | 30.55 | 0.177 | 86.16 | 29.17 |
| | | 1D-Partitioned SSDU | 0.083 | 94.07 | 35.59 | 0.114 | 88.99 | 32.85 | 0.134 | 84.75 | 31.41 | 0.148 | 81.61 | 30.55 |

Table 2: Accelerated MRI reconstruction performance evaluation using Normalized Root Mean Squared Error (NRMSE ↓), Structural Similarity Index Measure (SSIM ↑), and Peak Signal-to-Noise Ratio (PSNR ↑) averaged across 100 test samples.

## 3.2 ACCELERATED MRI RECONSTRUCTION

We now evaluate the accelerated reconstruction performance of Ambient Diffusion models trained on subsampled MRI data. Our method, *A-DPS* was implemented by following Eq. 2.11 for 500 steps with likelihood weighting $\gamma_t = \frac{1}{\|\boldsymbol{y}-A\mathbb{E}[\boldsymbol{x}_0|\boldsymbol{y}_{t,\text{train}}, A_{\text{train}}]\|_2}$. We consider the following baselines:

- **Fully-Sampled Diffusion Posterior Sampling (FS-DPS)**. For models trained on clean data, we can use the Diffusion Posterior Sampling (DPS) (Chung et al., 2023) algorithm. Specifically, we implemented the DPS algorithm with a Diffusion model trained with fully sampled ground truth images (FS-DPS). Inference was run using the update rule of Eq. 2.3, for a total of 500 steps and with $\gamma_t = \frac{1}{\|\boldsymbol{y}-A\mathbb{E}[\boldsymbol{x}_0|\boldsymbol{x}_t]\|_2}$.
- **Fully-Sampled Annealed Langevin Dynamics (FS-ALD)**. A popular method for solving inverse problems via score-based priors is ALD (Jalal et al., 2021). We train a score-based model using the same fully sampled $10,000$ samples as above and run inference for $1,300$ steps. This method acts as another fully sampled comparison to our technique.
- **L1-Wavelet Compressed Sensing (L1-CS)**. We use the same L1-Wavelet reconstruction described previously as a standalone non-deep learning comparison.
- **Model Based Deep Learning Supervised (MoDL-Sup)**. We use a supervised end-to-end model using the MoDL architecture in a supervised fashion (Aggarwal et al., 2019) for each acceleration factor. For inference, we pass the undersampled data through the trained MoDL network. Further details of this baseline are mentioned in Section C in the appendix.
- **Ambient Annealed Langevin Dynamics (A-ALD)**. We utilize our Ambient Diffusion MRI models with the inverse problem solver ALD (Jalal et al., 2021) for $1,300$ steps.
- **Ambient One Step (A-OS)**. Our method also admits a one-step solution. This is outlined in section 2.4 by noting that we train our model to estimate $\mathbb{E}[\boldsymbol{x}_0|\tilde{\boldsymbol{y}}_{\text{train}}, \tilde{A}_{\text{train}}]$. Thus we can use our models in a one-step fashion. The performance of the one-step prediction should only be expected to be good in-distribution, i.e. when the model is evaluated at the same acceleration factor as the one during training.
- **L1-Diffusion Posterior Sampling (L1-DPS)**. We utilize the diffusion models trained on the L1-Wavelet reconstructions of data from measurement sets at $R = 2, 4, 6, 8$. For reconstruction, we use DPS for a total of 500 steps with $\gamma_t = \frac{1}{\|\boldsymbol{y}-A\mathbb{E}[\boldsymbol{x}_0|\boldsymbol{x}_t]\|_2}$.
- **Self-Supervised learning via Data Under-sampling (SSDU)**. We train SSDU (Yaman et al., 2020) models using the end-to-end restoration network by splitting the available

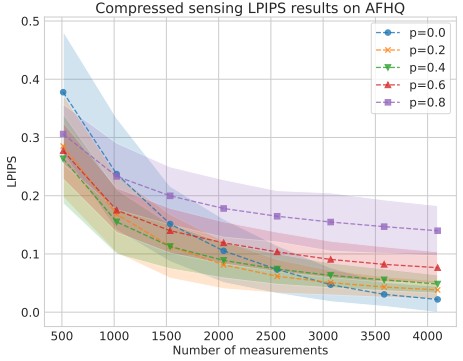 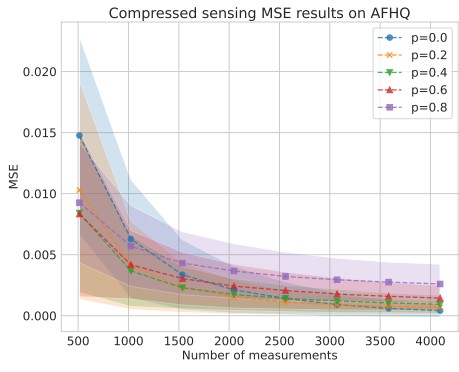

(a) LPIPS per Number of Measurements.    (b) MSE per Number of Measurements.

Figure 5: Compressed Sensing results, AFHQ: performance metric and standard deviation. As shown, the model trained with clean data ($p = 0.0$) only outperforms the models trained with corrupted data for more than $1000$ measurements, in both LPIPS and MSE.

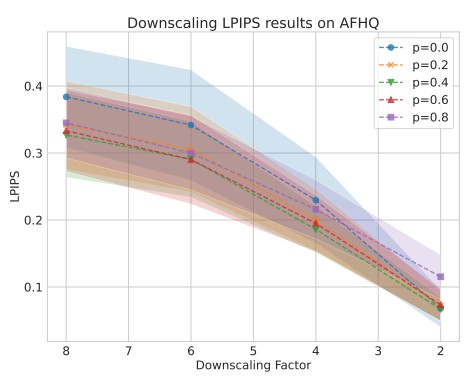 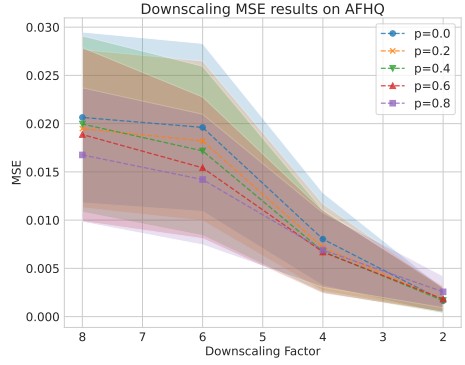

(a) LPIPS per downsampling factor.    (b) MSE per downsampling factor.

Figure 6: Super-resolution results, AFHQ: Performance metric and standard deviation. The model trained with clean data ($p = 0.0$) performs worse, except at downscaling factor 2.

measurements ($\Omega$) of each sample $y_i^\Omega$ into training sets ($\Theta$) and loss sets ($\Lambda$) or measurements $y_i^\Theta$ and $y_i^\Lambda$. Further SSDU training details are given in Section C.

- **1D-Partitioned SSDU**. We train 1D-Partitioned SSDU (Millard & Chiew, 2023) which stems from the SSDU framework and introduces partitioning of the sampling set to ensure each subset has a distribution similar to the original mask. The loss function, network architecture, and training hyperparameters were set according to Millard & Chiew (2023).

**Results** Quantitative metrics for the listed methods are included in Table 2 under the NRMSE, SSIM and PSNR metrics respectively. Furthermore, we report feature-based metrics including LPIPS and DISTS in Table 3. At low acceleration levels ($R = 2, 4$) FS-DPS outperforms most methods, including A-DPS. However, as the validation acceleration increases to higher ratios $R$, A-DPS outperforms other models, including those trained on fully sampled data.

We also trained an Ambient Diffusion model on undersampled abdominal MRI scans, where no ground truth data is available (Zhang et al., 2013). We show visual examples from the prior and posterior sampling experiments in the Appendix (Figs 7, 8).

We visualize reconstructions for a subset of the methods listed at various accelerations in Fig. 3. Here, we see that FS-DPS outperforms A-DPS at lower accelerations ($R = 4$). However, at higher acceleration ($R = 8$) we can see that FS-DPS introduces significant artifacts into the reconstruction while our A-DPS reconstructions maintain a high degree of visual fidelity. We also report the

distribution metrics (Conditional FID) between samples from the posterior and the ground-truth distribution for A-DPS and FS-DPS methods in Table 1. Similar to our findings from posterior reconstruction metrics, we find that in the higher acceleration ($R = 8$) regime, A-DPS outperforms FS-DPS. We also include graphical trends for NRMSE, SSIM, and PSNR for A-DPS, A-OS, FS-DPS, L1-DPS, FS-ALD, L1-CS, MoDL-Sup, and SSDU in the Appendix (Figs 9, 10, 11, 12, 13).

### 3.3 AMBIENT DIFFUSION WITH PRE-TRAINED MODELS ON DATASETS OF NATURAL IMAGES

We use the models from the Ambient Diffusion (Daras et al., 2023b) that are trained on randomly inpainted data with different erasure probabilities. Specifically, for AFHQ we use the Ambient Models with erasure probability $p \in \{0.2, 0.4, 0.6, 0.8\}$ and for Celeb-A we use the pre-trained models with $p \in \{0.6, 0.8, 0.9\}$. We underline that all the Ambient Models have worse performance for unconditional generation than those trained with clean data (i.e. the models trained with $p = 0.0$). This work aims to explore the conditional generation performance of Ambient Models, where the conditioning is in the measurements $\boldsymbol{y}_{\text{inf}}$, and compare it with models trained on uncorrupted data. To ensure that Ambient Models do not have an unfair advantage, we test only on restoration tasks that differ from those encountered in their training. Specifically, we use models trained on random inpainting and we evaluate Gaussian Compressed Sensing (Baraniuk, 2007) and Super Resolution.

**Results.**    In Figure 4 we show visual examples from our experiment on the noisy super-resolution task for natural images using a model: (1) trained with fully-sampled data (FS-DPS), (2) trained with randomly inpainted data at $p = 0.6$ (A-DPS). Figure 5 presents Gaussian Compressed Sensing reconstruction results (i.e. reconstructing a signal from Gaussian random projections). We show MSE and LPIPS performance metrics for the AFHQ dataset across varying number of measurements. The results are given for models that are *trained with inpainted* images at different levels of corruption, indicated by the erasure probability $p$. The model trained with clean data outperforms the models trained with corrupted data when the number of measurements is high. However, as we reduce the number of measurements, Ambient Models outperform the models trained with clean data in the very low measurements regime. To the best of our knowledge, there is no known theoretical argument that explains this performance cross-over, and understanding this further is an interesting research direction. Similar results are presented in Figure 6 for the task of super-resolution at AFHQ. The model trained on clean data ($p = 0$) slightly outperforms the Ambient Models in both LPIPS and MSE for reconstructing a $2\times$ downsampled image, as expected. Yet, as the resolution decreases, there is again a cross-over in performance and models trained on corrupted data start to outperform the models trained on uncorrupted data. We include results for LPIPS and MSE for Compressed Sensing and Downsampling in FFHQ and Celeb-A in the Appendix (Figs 15, 16, 17, 18).

We compute MSE in other inverse problem settings for FFHQ: Box Inpainting, Additive Gaussian Noise, and Super Resolution (with Gaussian Noise $\sigma = 0.05$). We find that trends in this experiment are consistent with previous findings. The results across multiple downsampling factors and noise levels in the Appendix (Figs 19, 20, 21). Finally, we examine how the number of sampling steps affects the performance. The MSE results for Compressed Sensing with $4000$ measurements on AFHQ are shown in Figure 14. As shown, the higher the erasure probability $p$ during training, the better the Compressed Sensing performance of the model for low Number of Function Evaluations (NFEs). Models trained with higher corruption are faster since they require fewer steps for similar performance. For increased NFEs, the models trained on clean(er) data outperform. This result is consistent across different datasets (AFHQ, FFHQ, CelebA), reconstruction tasks (Compressed Sensing, Downsampling), and metrics (MSE, LPIPS) (Figs 22, 23, 24, 25, 26 in the Appendix).

## 4    CONCLUSION

We present Ambient Diffusion Posterior Sampling (A-DPS), a simple framework based on DPS for solving inverse problems with Ambient Diffusion models. We show that Ambient Diffusion models trained on corrupted data can be better suited for handling ill-posed inverse problems under severe corruption. By being exposed to corrupted training data, A-DPS exhibits robust priors that generalize better to high-acceleration regimes, unlike FS-DPS, which may overfit clean data and fail to adapt effectively to severe undersampling. Our framework fully unlocks the potential of Ambient Diffusion models that are critical in applications where access to full data is impossible or undesirable.

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

# A  THEORETICAL RESULTS

## A.1  THEORY PROOF

In this section, we provide the proof of Theorem 2.1 that was stated in the main paper. We begin by giving the formal statement of the theorem.

**Theorem A.1** (Formal Statement of Theorem 2.1). *Let $\boldsymbol{x} \in \mathbb{C}^n$ be an unknown signal from a distribution that admits density $p(\boldsymbol{x})$. Let $\{S_i\}$ be a random set of diagonal matrices $\in \mathbb{C}^n$ that satisfies: $\sum_i S_i^H S_i = I_{n \times n}$. Assume sample access to linearly corrupted measurements of $\boldsymbol{x}$ given by:*

$$\boldsymbol{y}_{\text{train}} = \underbrace{\left(\sum_i S_i^H \mathcal{F}^{-1} P \mathcal{F} S_i\right)}_{A_{\text{train}}} \boldsymbol{x}, \tag{A.1}$$

*where $\mathcal{F}$ is the discrete Fourier matrix and $P$ is a random inpainting matrix, i.e. a diagonal matrix with either zeros or ones in the diagonal, such that $\Pr[P_{ii} = 1] > 0$, for all entries $i$ of the diagonal. Define:*

$$\tilde{\boldsymbol{y}}_{\text{train}} = \underbrace{\left(\sum_i S_i^H \mathcal{F}^{-1} \tilde{P} \mathcal{F} S_i\right)}_{\tilde{A}_{\text{train}}} \boldsymbol{x}, \tag{A.2}$$

*where $\tilde{P}$ is a further corrupted version of $P$ in the sense that with some non-zero probability $p$, a diagonal element that was 1 in $P$ becomes 0 in $\tilde{P}$. Define also $\boldsymbol{x}_t, \boldsymbol{y}_{t,\text{train}}, \tilde{\boldsymbol{y}}_{t,\text{train}}$ the noisy versions of $\boldsymbol{x}, \boldsymbol{y}_{\text{train}}, \tilde{\boldsymbol{y}}_{\text{train}}$ respectively, as in:*

$$\boldsymbol{x}_t = \boldsymbol{x} + \sigma_t \boldsymbol{\eta}, \quad \boldsymbol{y}_{t,\text{train}} = A_{\text{train}} \left(\boldsymbol{x} + \sigma_t \boldsymbol{\eta}\right), \quad \tilde{\boldsymbol{y}}_{t,\text{train}} = \tilde{A}_{\text{train}} \left(\boldsymbol{x} + \sigma_t \boldsymbol{\eta}\right). \tag{A.3}$$

*Then, the minimizer of the objective:*

$$J(\theta) = \mathbb{E}_{\boldsymbol{y}_{0,\text{train}}, \tilde{\boldsymbol{y}}_{t,\text{train}}, A_{\text{train}}, \tilde{P}} \left[\left|\left|A_{\text{train}} \boldsymbol{h}_\theta(\tilde{\boldsymbol{y}}_{t,\text{train}}, \tilde{P}) - \boldsymbol{y}_{0,\text{train}}\right|\right|^2\right], \tag{A.4}$$

*is: $\boldsymbol{h}_\theta(\tilde{\boldsymbol{y}}_{t,\text{train}}, \tilde{P}) = \mathbb{E}[\boldsymbol{x}_0|\tilde{\boldsymbol{y}}_{t,\text{train}}, \tilde{P}]$.*

*Proof.* We adapt the proof Theorem 4.1 in Ambient Diffusion (Daras et al., 2023b). To avoid notation clutter, we will be omitting the subscript "train", when necessary.

Let $\boldsymbol{h}_{\theta^*}(\tilde{\boldsymbol{y}}_t, \tilde{P}) = \mathbb{E}[\boldsymbol{x}_0|\tilde{\boldsymbol{y}}_t, \tilde{P}] + \boldsymbol{f}(\tilde{\boldsymbol{y}}_t, \tilde{P})$ be the optimal solution. The value of the objective for the optimal solution becomes:

$$J(\theta^*) = \mathbb{E}_{\boldsymbol{y}_0, \tilde{\boldsymbol{y}}_t, A, \tilde{P}} \left[\left|\left|A\mathbb{E}[\boldsymbol{x}_0|\tilde{\boldsymbol{y}}_t, \tilde{P}] + A\boldsymbol{f}(\tilde{\boldsymbol{y}}_t, \tilde{P}) - \boldsymbol{y}_0\right|\right|^2\right] \tag{A.5}$$

$$= \mathbb{E}_{\boldsymbol{y}_0, \tilde{\boldsymbol{y}}_t, A, \tilde{P}} \left[\underbrace{\left|\left|A\mathbb{E}[\boldsymbol{x}_0|\tilde{\boldsymbol{y}}_t, \tilde{P}] - \boldsymbol{y}_0\right|\right|^2}_{\text{irreducible error}} + \boldsymbol{f}(\tilde{\boldsymbol{y}}_t, \tilde{P})^T A^T A \boldsymbol{f}(\tilde{\boldsymbol{y}}_t, \tilde{P}) - 2\left(\mathbb{E}[\boldsymbol{x}_0|\tilde{\boldsymbol{y}}_t, \tilde{P}] - \boldsymbol{x}_0\right)^T A^T A \boldsymbol{f}(\tilde{\boldsymbol{y}}_t, \tilde{P})\right]. \tag{A.6}$$

We will now work with the last term.

$$\mathbb{E}_{\boldsymbol{x}_0, \tilde{\boldsymbol{y}}_t, A, \tilde{P}} \left[\left(\mathbb{E}[\boldsymbol{x}_0|\tilde{\boldsymbol{y}}_t, \tilde{P}] - \boldsymbol{x}_0\right)^T A^T A \boldsymbol{f}(\tilde{\boldsymbol{y}}_t, \tilde{P})\right] \tag{A.7}$$

$$= \mathbb{E}_{\tilde{\boldsymbol{y}}_t, A, \tilde{P}} \left[\mathbb{E}_{\boldsymbol{x}_0|\tilde{\boldsymbol{y}}_t, A, \tilde{P}} \left[\left(\mathbb{E}[\boldsymbol{x}_0|\tilde{\boldsymbol{y}}_t, \tilde{P}] - \boldsymbol{x}_0\right)^T A^T A \boldsymbol{f}(\tilde{\boldsymbol{y}}_t, \tilde{P})\right]\right] \tag{A.8}$$

$$= \mathbb{E}_{\tilde{\boldsymbol{y}}_t, A, \tilde{P}} \left[\left(\mathbb{E}[\boldsymbol{x}_0|\tilde{\boldsymbol{y}}_t, \tilde{P}] - \mathbb{E}[\boldsymbol{x}_0|\tilde{\boldsymbol{y}}_t, A, \tilde{P}]\right)^T A^T A \boldsymbol{f}(\tilde{\boldsymbol{y}}_t, \tilde{P})\right] \tag{A.9}$$

$$= 0, \tag{A.10}$$

since $\mathbb{E}[\boldsymbol{x}_0|\tilde{\boldsymbol{y}}_t, \tilde{P}] = \mathbb{E}[\boldsymbol{x}_0|\tilde{\boldsymbol{y}}_t, A, \tilde{P}]$, i.e. the value of $\boldsymbol{x}_0$ does not depend on $\tilde{A}$ given $\tilde{P}$ and $\tilde{\boldsymbol{y}}_t$.

We will now work with the middle term. We have that:

$$\mathbb{E}_{\tilde{\boldsymbol{y}}_t, A, \tilde{P}}\left[\boldsymbol{f}(\tilde{\boldsymbol{y}}_t, \tilde{P})^T A^T A \boldsymbol{f}(\tilde{\boldsymbol{y}}_t, \tilde{P})\right] \tag{A.11}$$

$$= \mathbb{E}_{\tilde{\boldsymbol{y}}_t, \tilde{P}}\left[\mathbb{E}_{A|\tilde{\boldsymbol{y}}_t, \tilde{P}}\left[\boldsymbol{f}(\tilde{\boldsymbol{y}}_t, \tilde{P})^T A^T A \boldsymbol{f}(\tilde{\boldsymbol{y}}_t, \tilde{P})\right]\right] \tag{A.12}$$

$$= \mathbb{E}_{\tilde{\boldsymbol{y}}_t, \tilde{P}}\left[\boldsymbol{f}(\tilde{\boldsymbol{y}}_t, \tilde{P})^T \mathbb{E}[A^T A|\tilde{P}]\boldsymbol{f}(\tilde{\boldsymbol{y}}_t, \tilde{P})\right]. \tag{A.13}$$

From the last equation, it is evident that if $\mathbb{E}[A^T A|\tilde{P}]$ is full-rank, and hence the minimizer is $\mathbb{E}[\boldsymbol{x}_0|\tilde{\boldsymbol{y}}_t, \tilde{P}]$ as needed. Since our $A$ matrix is square, it suffices to show that $\mathbb{E}[A|\tilde{P}]$ is full-rank. By Corollary A.1. in Ambient Diffusion, we have that: $\mathbb{E}[P|\tilde{P}]$ is full-rank. We will now show that $\mathcal{F}^{-1}\mathbb{E}[P|\tilde{P}]\mathcal{F}$ is also full-rank. This can be easily proved with contradiction. Assume that $\mathcal{F}^{-1}\mathbb{E}[P|\tilde{P}]\mathcal{F}$ is not full-rank. Then, there exists a vector $\boldsymbol{w} \neq 0$ such that:

$$\mathcal{F}^{-1}\mathbb{E}[P|\tilde{P}]\mathcal{F}\boldsymbol{w} = 0 \iff \tag{A.14}$$

$$\mathbb{E}[P|\tilde{P}]\underbrace{\mathcal{F}\boldsymbol{w}}_{\boldsymbol{z} \neq \boldsymbol{0}} = 0 \iff \tag{A.15}$$

$$\mathbb{E}[P|\tilde{P}]\boldsymbol{z} = 0, \tag{A.16}$$

which is a contradiction, since $\boldsymbol{z} \neq \boldsymbol{0}$ and $\mathbb{E}[P|\tilde{P}]$ is full-rank.

So far, we have established that $\mathcal{F}^{-1}\mathbb{E}[P|\tilde{P}]\mathcal{F}$ is full-rank. By linearity of expectation, we further have that:

$$\mathcal{F}^{-1}\mathbb{E}[P|\tilde{P}]\mathcal{F} = \mathbb{E}[\mathcal{F}^{-1}P\mathcal{F}|\tilde{P}], \tag{A.17}$$

and hence, the latter is also full-rank. Finally, since $\sum_i S_i^H S_i = I$, for any full-rank matrix $C$, we have that $\sum_i S_i^H C S_i$ is also full-rank. This is true for any given set $S_i$ and hence it is also true for the expectation over the sets $\{S_i\}$. Putting everything together, we have that:

$$\mathbb{E}_{\{S_i\}, P}\left[\sum_i S_i^H \mathcal{F}^{-1}\mathbb{E}[P|\tilde{P}]\mathcal{F}S_i \,\middle|\, \tilde{P}\right] = \mathbb{E}[A|\tilde{P}], \tag{A.18}$$

is full-rank and the proof is complete.

$\square$

**Note on alternative Ambient Diffusion designs.** Our Ambient Diffusion MRI training approach can be extended to use other linear operators to aggregate the measurements from different coils (e.g. we could have used the pseudoinverse). Alternatively, we can also further condition on the sensitivity maps, or directly feed all the coil measurements as input to the network without aggregation. We opted for a simple and fast method (aggregation through the adjoint). However, these other approaches can lead to improved performance because they fully leverage coil information.

## B  DATASET PREPARATION

We follow standard practices in the MRI literature. We download the FastMRI dataset, randomly select 2,000 T2-Weighted scans, and pick 5 center slices from each scan, giving us us a dataset of 10,000 T2-weighted brain scans ($k$-space measurements). We pass each multi-channel $k$-space sample through a noise pre-whitening filter to transform the noise into standard white Gaussian. Then, we normalize each $k$-space sample by the absolute value of 99-th percentile of the root sum-of-squares reconstruction of the autocalibration region ($24x \times 24$ pixels from the center). Given the fully sampled $k$-space samples, we estimate sensitivity maps using the ESPIRiT calibration method (Uecker et al., 2014). These measurements serve as a fully sampled reference dataset (i.e., $R = 1$).

## C   DETAILS FOR BASELINES.

**L1 Methods.**   We used the BART implementation (Uecker et al., 2016; Blumenthal et al., 2022) and searched for the best regularization parameter over the training set. Specifically, we used a regularization weighting of 0.001 with 100 iterations.

**Self-Supervised learning via Data Under-sampling (SSDU) (Yaman et al., 2020)**   In a supervised setting, end-to-end image restoration networks can be trained by passing measurements through a network like MoDL (Aggarwal et al., 2019) or VarNet (Hammernik et al., 2018) and taking a loss directly on the output of the network with the ground truth image. In the self-supervised setting, however, this is not possible. SSDU, trains the end-to-end restoration network by splitting the available measurement set ($\Omega$) of each sample $y_i^\Omega$ into training sets ($\Theta$) and loss sets ($\Lambda$) or measurements $y_i^\Theta$ and $y_i^\Lambda$ respectively where $\Omega = \Theta \cup \Lambda$. Where the reconstruction network is given access to measurements in the training set to obtain an estimated image $\hat{\boldsymbol{x}}_i^\Theta = \boldsymbol{h}_\theta(y_i^\Theta)$ and the loss is then defined over the loss measurements as

$$L(\boldsymbol{y}_i^\Lambda, \hat{\boldsymbol{x}}_i^\Theta) = \frac{\left\| \boldsymbol{y}_i^\Lambda - A_\Lambda \hat{\boldsymbol{x}}_i^\Theta \right\|_1}{\left\| \boldsymbol{y}_i^\Lambda \right\|_1} + \frac{\left\| \boldsymbol{y}_i^\Lambda - A_\Lambda \hat{\boldsymbol{x}}_i^\Theta \right\|_2}{\left\| \boldsymbol{y}_i^\Lambda \right\|_2} \qquad \text{(C.1)}$$

where $A_\Lambda$ is the forward operator with ones in the selection mask $P$ at only the measurement locations in the loss set and zeros elsewhere. In this work, we selected $\boldsymbol{h}_\theta(\cdot)$ to have the MoDL architecture (Aggarwal et al., 2019). We trained individual models at four different acceleration levels ($R = 2, 4, 6, 8$). Each model was trained for ten epochs using the same training set ($10,000$ slices) as previously described. We used uniform random sampling to separate the training and loss measurement groups for each sample and found that a split of $\rho = \frac{|\Lambda|}{|\Omega|} = 0.2$ provided the best performance and thus was used for reporting all subsequent metrics.

**MoDL Supervised (MoDL-Sup)**   To provide an upper bound on performance for supervised end-to-end methods we also trained the MoDL architecture in a supervised fashion (Aggarwal et al., 2019). Again, we trained individual models at four different acceleration levels ($R = 2, 4, 6, 8$). Each model was trained using the same $10,000$ slices as above. However, for these models, we used the normalized root mean squared error (NRMSE) as the loss function:

$$L(\boldsymbol{x}_i, \hat{\boldsymbol{x}}_i) = \frac{\left\| \boldsymbol{x}_i - \hat{\boldsymbol{x}}_i \right\|_2}{\left\| \boldsymbol{x}_i \right\|_2} \qquad \text{(C.2)}$$

where $\boldsymbol{x}_i$ are the ground truth images from our dataset and $\hat{\boldsymbol{x}}_i = \boldsymbol{h}_\theta(\boldsymbol{y}_i, A_i)$ are the reconstructions provided by our network based on under-sampled measurements $\boldsymbol{y}_i$.

## D   HYPERPARAMETERS FOR MRI MODELS

**Training Hyperparameters.**   The EDM model trained on the fully-sampled ($R = 1$) dataset, as well as the four EDM models trained on L1-Wavelet reconstructions at each acceleration factor $R \in \{2, 4, 6, 8\}$ were all trained with 65 million parameters. The Ambient Diffusion models on the other hand were trained with 36 million parameters, for faster training. All the Ambient Diffusion models were trained for 250 epochs. For the Ambient Diffusion training experiments, the further corrupted sampling masks are given as an input to the model by concatenating with the measurements along the channel dimension, as in the original Ambient Diffusion (Daras et al., 2023b) paper.

**Sampling Hyperparameters.**   The only tunable parameters for DPS (Eq. 2.3) and Ambient DPS (Eq. 2.11) are in the scheduling of the magnitude of the measurements likelihood term. In all the experiments in the DPS paper, this term is kept constant throughout the diffusion sampling trajectory and the authors recommend selecting a value in the range between $[0.1, 10]$. We follow this recommendation and we keep this term constant. The value of the step size for each model is selected with a hyperparameter search in the recommended range. For all our experiments, we follow exactly the DPS implementation provided in the official code repository of the paper. The other parameter

that impacts performance is the number of steps we are going to run each algorithm for, i.e. the discretization level of the SDEs of Equations 2.3, 2.11. Typically, the higher the number of steps the better the performance since the discretization error decreases (Chen et al., 2022b; 2023). For the performance results, we run each method for several steps $\in \{50, 100, 150, 200, 250, 300\}$.

# E ADDITIONAL MRI RESULTS

Table 3: Accelerated MRI reconstruction performance evaluation using LPIPS ↓ and DISTS ↓, averaged across 100 test samples.

| Training Method | Training Data | Reconstruction Method | $R = 2$ | | $R = 4$ | | $R = 6$ | | $R = 8$ | |
|---|---|---|---|---|---|---|---|---|---|---|
| | | | LPIPS | DISTS | LPIPS | DISTS | LPIPS | DISTS | LPIPS | DISTS |
| Supervised | $R = 1$ | FS-DPS | 0.061 | 0.028 | 0.106 | 0.059 | 0.166 | 0.094 | 0.231 | 0.126 |
| Self-Supervised | $R = 2$ | A-DPS | 0.118 | 0.062 | **0.127** | **0.068** | **0.138** | **0.076** | **0.150** | **0.082** |
| | | A-ALD | 0.119 | 0.063 | **0.127** | **0.068** | 0.139 | 0.077 | 0.151 | 0.083 |
| | | L1-DPS | **0.067** | **0.032** | 0.131 | 0.075 | 0.201 | 0.115 | 0.262 | 0.143 |
| | | 1D-Partitioned SSDU | 0.141 | 0.084 | 0.144 | 0.081 | 0.187 | 0.108 | 0.223 | 0.129 |
| | $R = 4$ | A-DPS | 0.155 | 0.090 | **0.155** | **0.092** | **0.161** | **0.095** | **0.168** | **0.099** |
| | | A-ALD | 0.156 | 0.090 | **0.155** | **0.092** | **0.161** | **0.095** | **0.168** | **0.099** |
| | | L1-DPS | **0.096** | **0.051** | 0.193 | 0.112 | 0.258 | 0.147 | 0.301 | 0.167 |
| | | 1D-Partitioned SSDU | 0.169 | 0.105 | 0.169 | **0.092** | 0.187 | 0.104 | 0.214 | 0.123 |
| | $R = 6$ | A-DPS | 0.180 | 0.110 | 0.178 | 0.110 | **0.181** | **0.111** | **0.186** | **0.114** |
| | | A-ALD | 0.179 | 0.109 | 0.179 | 0.111 | **0.181** | **0.111** | **0.186** | **0.114** |
| | | L1-DPS | **0.112** | **0.061** | 0.221 | 0.128 | 0.282 | 0.161 | 0.318 | 0.179 |
| | | 1D-Partitioned SSDU | 0.143 | 0.092 | **0.169** | **0.102** | 0.196 | 0.117 | 0.222 | 0.133 |
| | $R = 8$ | A-DPS | 0.212 | 0.129 | 0.204 | 0.127 | **0.204** | **0.127** | **0.204** | **0.127** |
| | | A-ALD | 0.213 | 0.130 | 0.204 | 0.127 | **0.204** | **0.127** | **0.204** | **0.127** |
| | | L1-DPS | **0.114** | **0.064** | 0.227 | 0.134 | 0.291 | 0.168 | 0.327 | 0.187 |
| | | 1D-Partitioned SSDU | 0.142 | 0.094 | **0.176** | **0.109** | 0.206 | **0.127** | 0.231 | 0.140 |

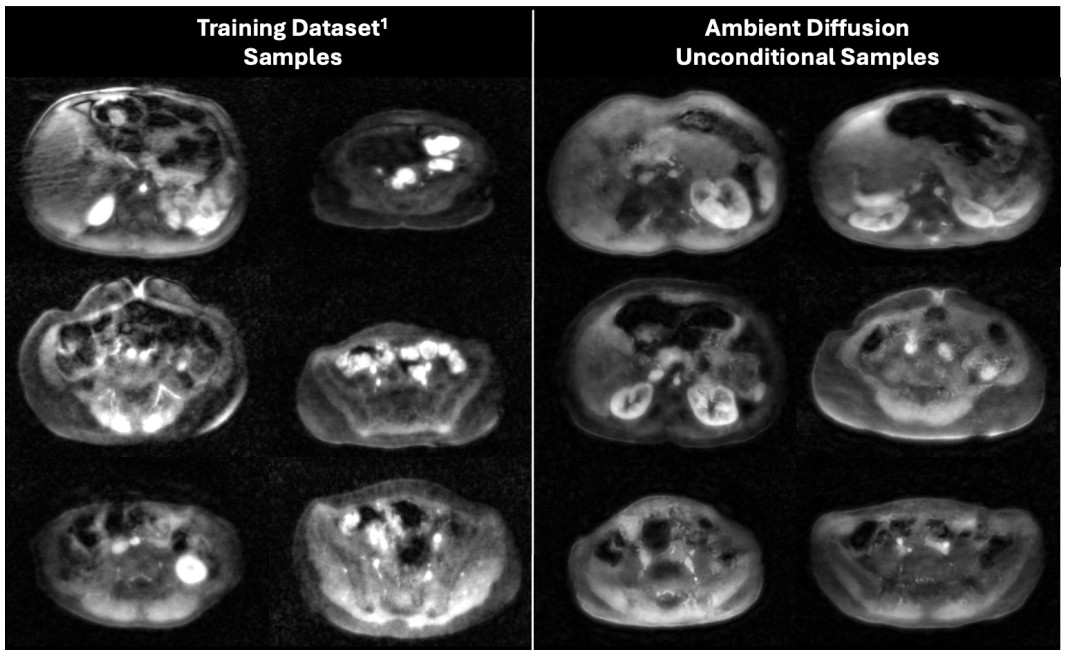

Figure 7: Training samples (left) and prior samples (right) from ambient diffusion model trained on undersampled abdominal MRI scans, where no ground truth data is available.

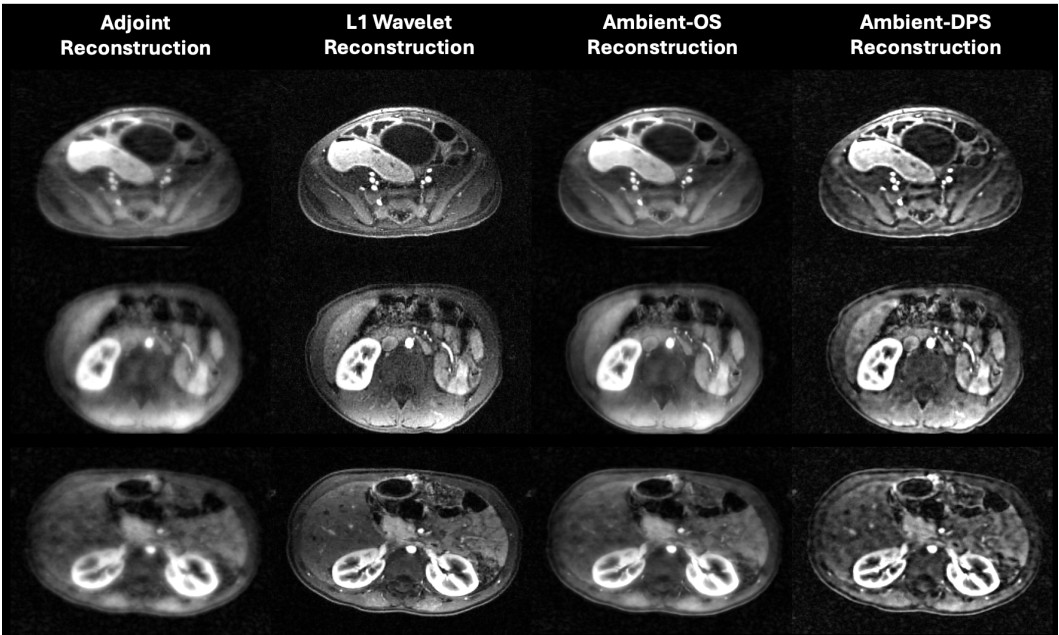

Figure 8: Ambient diffusion posterior sampling (A-DPS) performance using ambient diffusion model trained on undersampled abdominal MRI scans, where no ground truth data is available.

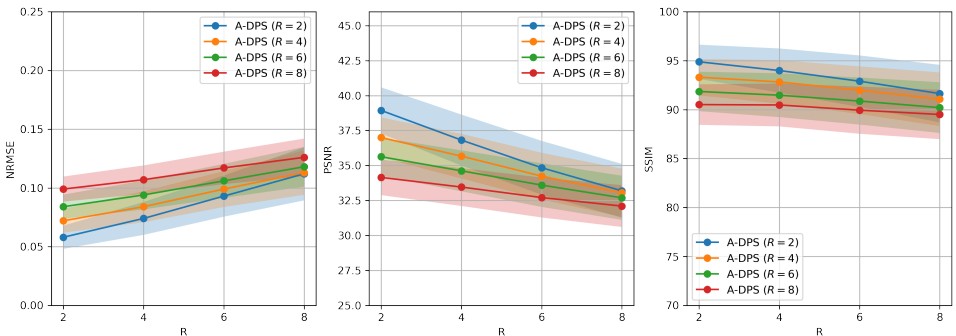

Figure 9: Ambient diffusion posterior sampling (A-DPS) multi-step performance metrics at $R = 2, 4, 6, 8$ for models trained at $R = 2, 4, 6, 8$.

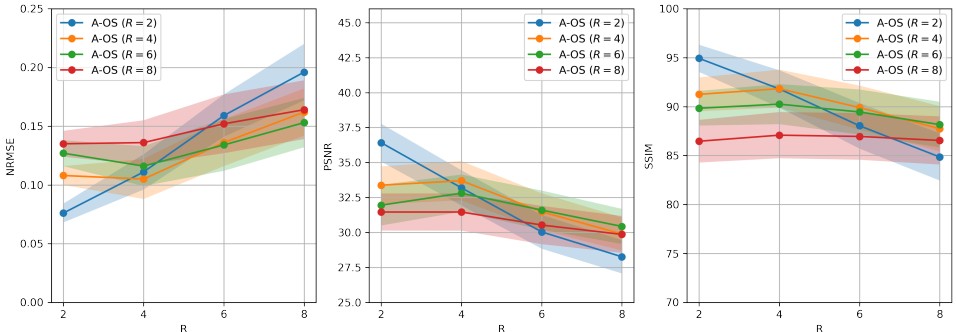

Figure 10: Ambient diffusion one step (A-OS) performance metrics at $R = 2, 4, 6, 8$ for models trained at $R = 2, 4, 6, 8$.

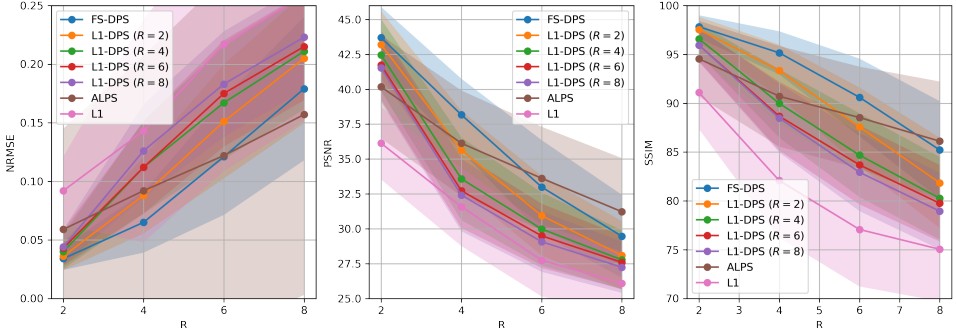

Figure 11: Performance metrics for FS-DPS, L1-DPS, FS-ALD, and L1-CS at $R = 2, 4, 6, 8$

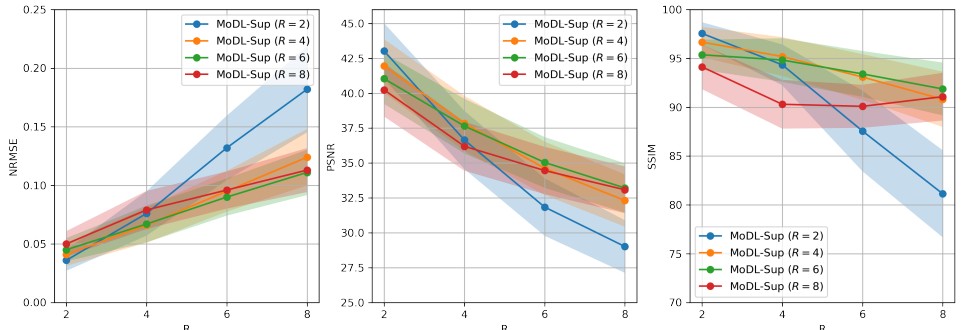

Figure 12: Fully supervised MoDL performance metrics at $R = 2, 4, 6, 8$ for models trained at $R = 2, 4, 6, 8$.

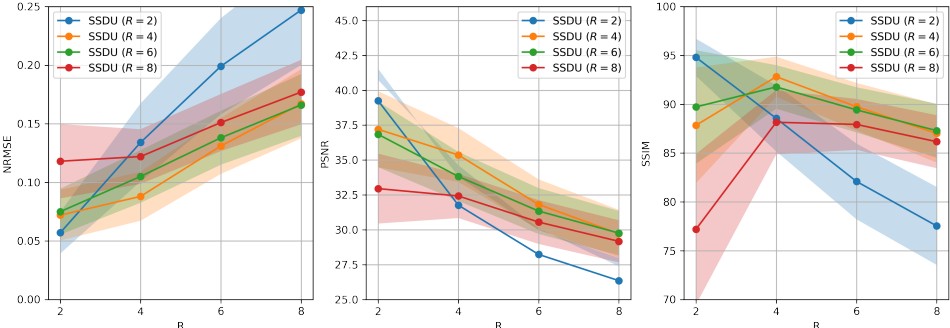

Figure 13: SSDU performance metrics at $R = 2, 4, 6, 8$ for models trained at $R = 2, 4, 6, 8$.

# F    ADDITIONAL NATURAL IMAGE PERFORMANCE RESULTS

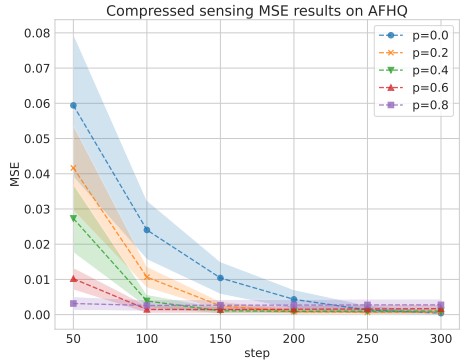

(a) Compressed Sensing with 4000 measurements per Number of Function Evaluations (NFEs).

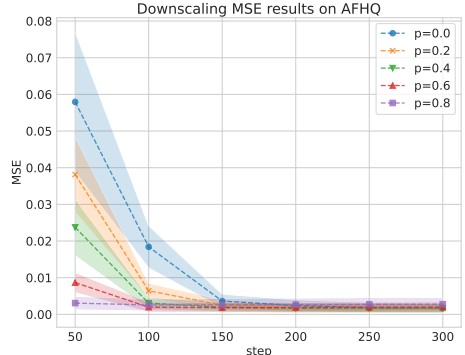

(b) 2× Super-Resolution per Number of Function Evaluations (NFEs).

Figure 14: Speed performance plots for AFHQ.

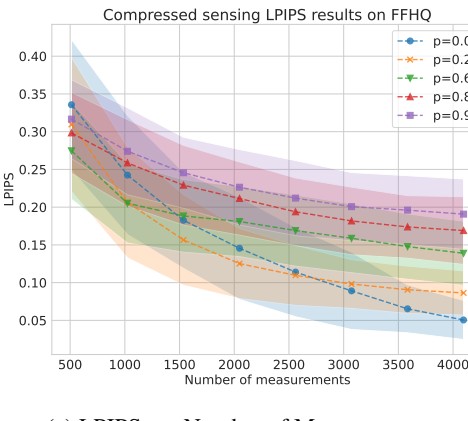

(a) LPIPS per Number of Measurements.

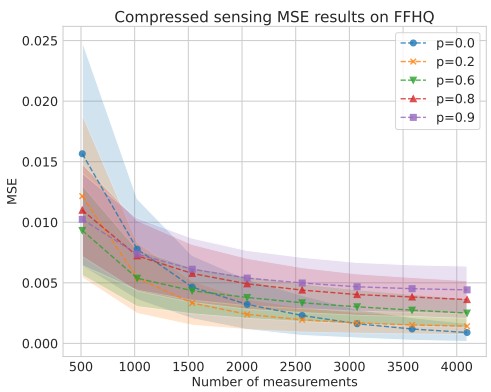

(b) MSE per Number of Measurements.

Figure 15: Compressed Sensing Results for FFHQ.

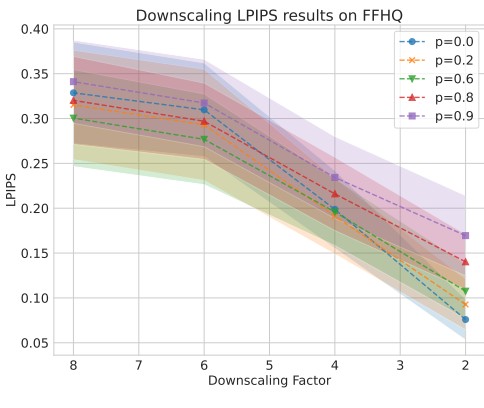

(a) LPIPS per Downscaling Factor.

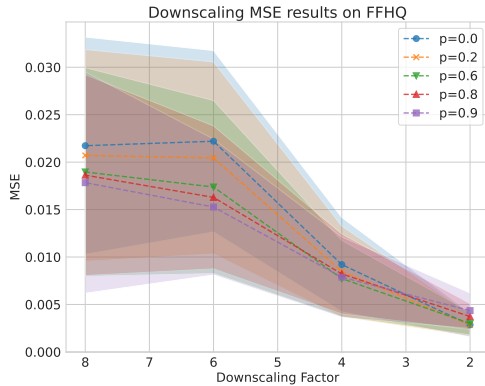

(b) MSE per Downscaling Factor.

Figure 16: Downscaling Results for FFHQ.

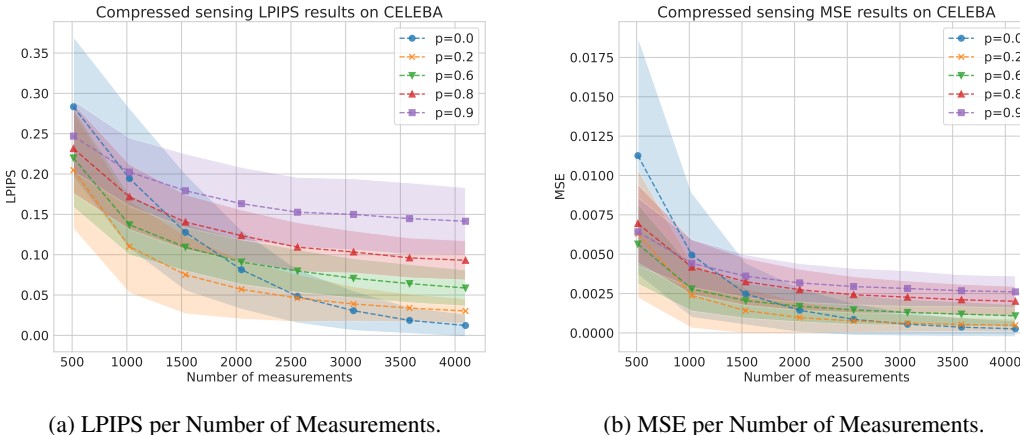

(a) LPIPS per Number of Measurements.

(b) MSE per Number of Measurements.

Figure 17: Compressed Sensing Results for Celeb-A.

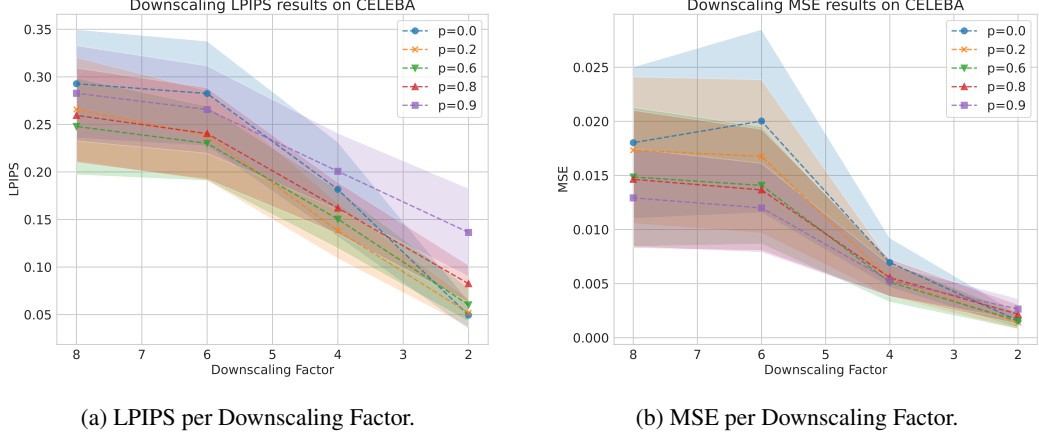

(a) LPIPS per Downscaling Factor.

(b) MSE per Downscaling Factor.

Figure 18: Downscaling Results for Celeb-A.

For the following experiments, we compare a model trained without corruption to a model trained with randomly inpainted data at $p = 0.6$. The evaluation dataset is FFHQ. The training size and hyperparameters for both models are the same. For both models, we extensively tune the sampling hyperparameters to maximize their performance (starting from the recommended hyperparameters from the DPS work). The finding we get in these additional experiments is that the model trained on corrupted data (inpainting) outperforms the model trained on clean data in the high corruption regime across all evaluation tasks.

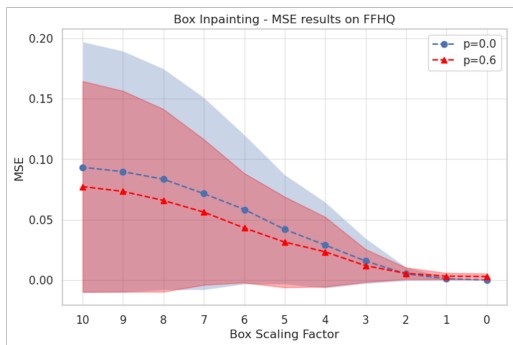

Figure 19: Box Inpainting results for FFHQ

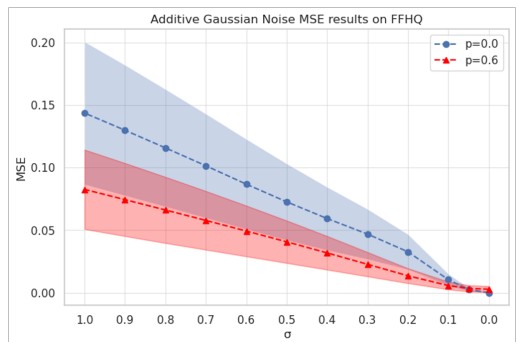

Figure 20: Additive Gaussian Noise results for FFHQ

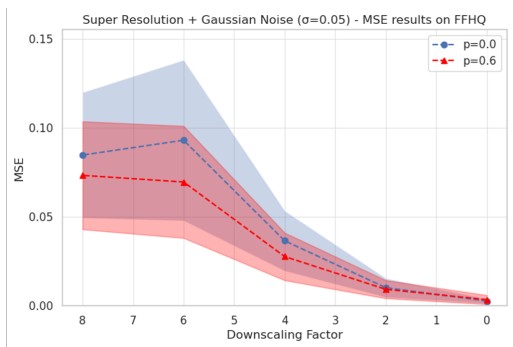

Figure 21: Super Resolution + Additive Gaussian Noise ($\sigma = 0.05$) results for FFHQ

## G    ADDITIONAL NATURAL IMAGE SPEED RESULTS

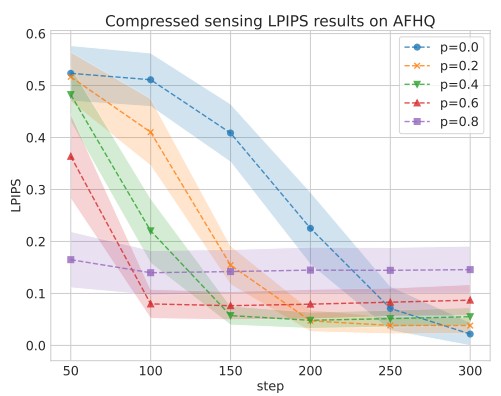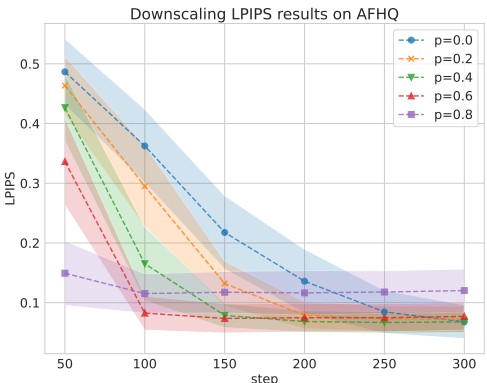

(a) Compressed Sensing with 4000 measurements per Number of Function Evaluations (NFEs).

(b) 2× Super-Resolution per Number of Function Evaluations (NFEs).

Figure 22: Speed LPIPS performance plots for AFHQ.

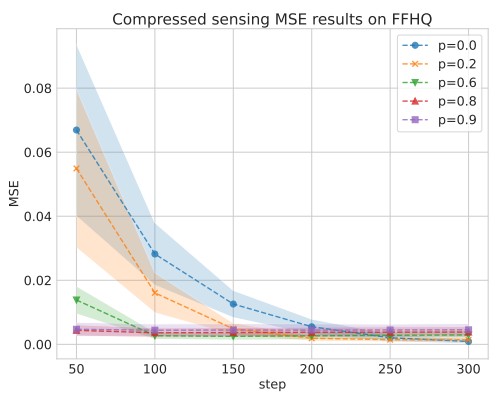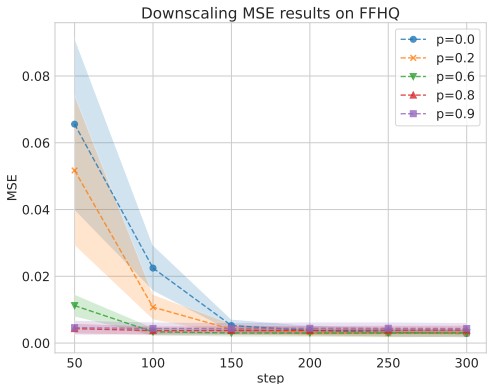

(a) Compressed Sensing with 4000 measurements per Number of Function Evaluations (NFEs).

(b) 2× Super-Resolution per Number of Function Evaluations (NFEs).

Figure 23: Speed MSE performance plots for FFHQ.

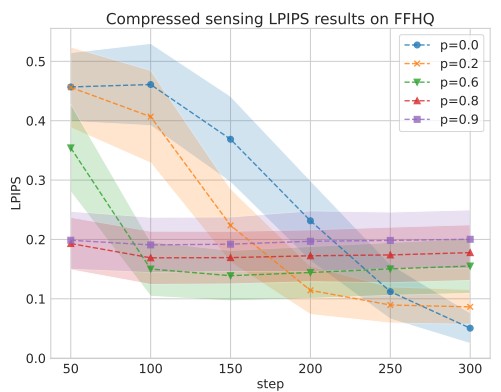
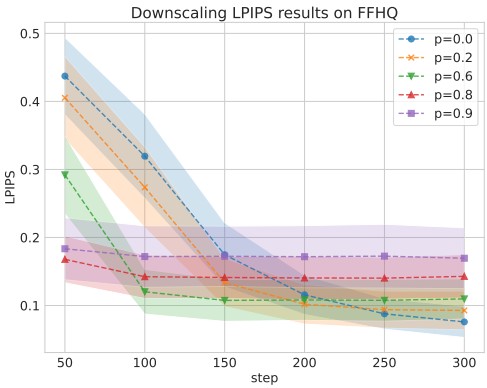

(a) Compressed Sensing with 4000 measurements per Number of Function Evaluations (NFEs).

(b) 2× Super-Resolution per Number of Function Evaluations (NFEs).

Figure 24: Speed LPIPS performance plots for FFHQ.

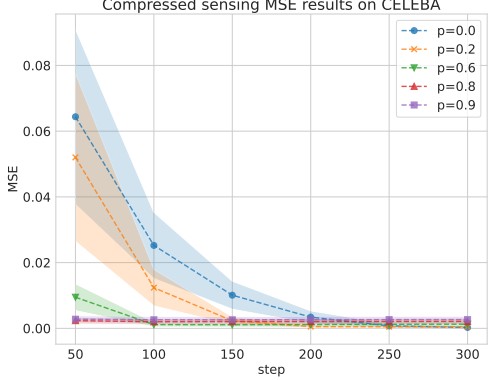
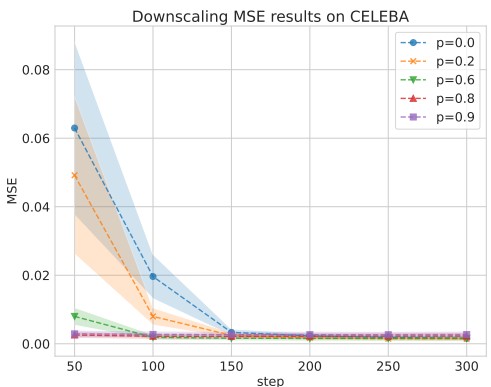

(a) Compressed Sensing with 4000 measurements per Number of Function Evaluations (NFEs).

(b) 2× Super-Resolution per Number of Function Evaluations (NFEs).

Figure 25: Speed MSE performance plots for Celeb-A.

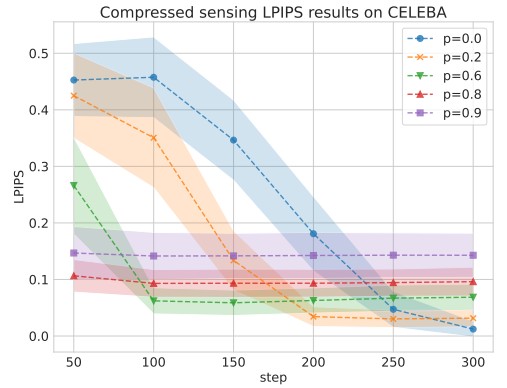 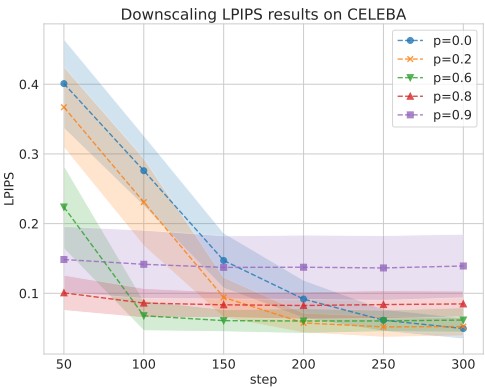

(a) Compressed Sensing with 4000 measurements per Number of Function Evaluations (NFEs).

(b) 2× Super-Resolution per Number of Function Evaluations (NFEs).

Figure 26: Speed LPIPS performance plots for Celeb-A.

## H  LIMITATIONS AND FUTURE DIRECTIONS.

Our work has several limitations. First, there is a lack of theoretical understanding of the observed experimental phenomenon: we do not have a good grasp of why models trained on highly corrupted data lead to better priors for inverse problems of another corruption type. Second, we only tested the reconstruction performance of our models using the DPS algorithm. Several other recent algorithms have been developed for solving inverse problems with diffusion models and it is unknown whether our findings generalize for these reconstruction algorithms. Finally, our method relies on the existence of Ambient Diffusion models. There are many more models available trained on clean data than models trained on highly corrupted data.

The auto-calibration signal (ACS) region is shared across training samples, which could lead to an over-representation of certain k-space regions. While our current approach does not explicitly address this, introducing a diagonal weighting matrix in the loss function, as suggested by Millard & Chiew (2023), is a promising direction to mitigate this issue.

Currently, we define $A_{train}$ according to Eq 2.7. This is convenient because it lets us use an image-to-image network architecture such as that used by EDM. An alternative could be to define $A_{train}$ directly as in Eq 2.5, in which case the network would go from multi-coil k-space to an image. This network could be preceded by an IFFT, which would effectively be a multi-coil-to-image network, and could be standardized using coil compression (Zhang et al., 2013). Such a network architecture could be interesting and potentially more expressive, as we currently collapse the multi-coil information through zero-filling adjoint. Another possibility is to define $A_{train}$ as the pseudo-inverse applied to $A$, which could be solved efficiently with the conjugate gradient algorithm.

## I  ACKNOWLEDGEMENTS

This research has been supported by NSF Grants CCF 1763702, NSF CCF-2239687 (CAREER), AF1901292, CNS 2148141, Tripods CCF 1934932, NSF AI Institute for Foundations of Machine Learning (IFML) 2019844, the Texas Advanced Computing Center (TACC) and research gifts by Western Digital, WNCG IAP, UT Austin Machine Learning Lab (MLL), Cisco and the Archie Straiton Endowed Faculty Fellowship. Giannis Daras has been supported by the Onassis Fellowship (Scholarship ID: F ZS 012-1/2022-2023), the Bodossaki Fellowship and the Leventis Fellowship.

