# OpenReview forum: "Ambient Diffusion Posterior Sampling: Solving Inverse Problems with Diffusion Models Trained on Corrupted Data"
_ICLR.cc/2025/Conference — ICLR 2025 Poster_

### Official Review · Reviewer_Q5u6 · 2024-10-19

**Soundness:** 4
**Presentation:** 3
**Contribution:** 4
**Rating:** 8
**Confidence:** 4

**Summary:**

In this work the authors provide a framework for solving inverse problems with diffusion models that are trained on corrupted data. The authors extend the existing ambient diffusion framework to enable Fourier domain-corrupted measurements and propose ambient diffusion posterior sampling, an algorithm that leverages generative models pre-trained on one type of corruption to solve other inverse problems. The authors show the performance of their method on accelerated MRI reconstruction and on various problems for natural images.

**Strengths:**

S1) The paper is very well written and provides an appropriate amount of detail. Similarly, all analysis is easy to follow.

S2) The authors include a nice theoretical result in Appendix A.

S3) The quantitative performance of A-DPS is impressive, and the selected baselines offer an easy way to contextualize it (particularly for MRI).

S4) In my opinion, and based on the results, this is the first diffusion-MRI method to truly push the state-of-the-art forward since [1].

[1] Ajil Jalal, Marius Arvinte, Giannis Daras, Eric Price, Alexandros G Dimakis, and Jon Tamir, "Robust
compressed sensing mri with deep generative priors," 2021

**Weaknesses:**

I find this paper to be very impactful (hence my score). That said, it is very important to me that the following two weaknesses are addressed:

W1) While the experimental performance of A-DPS is impressive, I think that the MRI results suffer from insufficient evaluation with all competitors. NRMSE being the only metric reported in Table 2 is a concern. It would be worthwhile to add some feature-based metrics like LPIPS/DISTS for all methods. You use VGG to compute FID, you can also use VGG for those metrics (since VGG agrees with radiologists' perceptions [1]). Alternatively, you could also use AlexNet as your feature extractor [1]. Either way, I think that it is important to see these results for your method and competitors. The lack of additional metrics holds this paper back. I understand the space concern given the size of Table 2, but even putting these results in an appendix would be useful to provide readers full context and make a convincing statement about A-DPS' superiority.

W2) I am confused by the bolding in Table 2. For methods trained with clean data, no values are bolded. For the R=2 and R=4 sections, A-DPS has equal performance to A-ALD in some metrics. The A-ALD metrics that are equally performant should be bolded for fairness. Please revise this table so that it is consistent.

[1] P. M. Adamson, A. D. Desai, J. Dominic, C. Bluethgen, J. P. Wood, A. B. Syed, R. D. Boutin, K. J. Stevens,
S. Vasanawala, J. M. Pauly, A. S. Chaudhari, and B. Gunel, “Using deep feature distances for evaluating
MR image reconstruction quality,” 2023

**Questions:**

See weaknesses. Although, I do have one actual question:

Q1) Did you consider training your MRI diffusion model (ambient or other) with the multi-coil data? To be clear: I am not asking you to do this, as it is a fundamental departure from what you have here. That said, I wonder if you tried this. I know that in, e.g., fastMRI, the number of coils is inconsistent, but you can get around this issue with virtual coils [1]. I suspect that the overall performance could be further improved with a multi-coil diffusion process, rather than a coil-combined one.

[1] T. Zhang, J. M. Pauly, S. S. Vasanawala, and M. Lustig, “Coil compression for accelerated imaging with
Cartesian sampling,” 2013

---

> ### Author Response · Authors · 2024-11-20
> **Response to Reviewer Q5u6**
>
> We thank a lot the Reviewer fo their valuable Review. We are glad that the Reviewer appreciated the novelty of our proposed algorithm and the overall presentation of our findings.
>
> **Feedback**: While the experimental performance of A-DPS is impressive, I think that the MRI results suffer from insufficient evaluation with all competitors. NRMSE being the only metric reported in Table 2 is a concern. It would be worthwhile to add some feature-based metrics like LPIPS/DISTS for all methods. You use VGG to compute FID, you can also use VGG for those metrics (since VGG agrees with radiologists' perceptions [1]). Alternatively, you could also use AlexNet as your feature extractor [1]. Either way, I think that it is important to see these results for your method and competitors. The lack of additional metrics holds this paper back. I understand the space concern given the size of Table 2, but even putting these results in an appendix would be useful to provide readers full context and make a convincing statement about A-DPS' superiority.
>
> **Response**: We have expanded our evaluation to include LPIPS and DISTS metrics, which are now reported in Table 3. Additionally, Table 2 has been updated to include SSIM and PSNR alongside NRMSE.  Regarding insufficient evaluation, we have implemented another self-supervised baseline stemming from SSDU [A]. Here are the results from this baseline. Similar to our previous results, we show that A-DPS outperforms other algorithms at high acceleration rates.
>
> [A] Millard, Charles, and Mark Chiew. "A theoretical framework for self-supervised MR image reconstruction using sub-sampling via variable density Noisier2Noise." IEEE Transactions on Computational Imaging (2023).
>
> Below we report NRMSE, LPIPS, and DISTS across 100 validation samples:
>
> | **Training Data** | **Reconstruction Method**       | **R=2** | **R=4** | **R=6** | **R=8** |
> |-------------------|---------------------------------|---------|---------|---------|---------|
> | R=1              | FS-DPS                          | 0.034   | 0.065   | 0.121   | 0.179   |
> | R=2              | A-DPS                           | 0.058   | 0.074   | 0.093   | 0.112   |
> | R=2              | **1D-Partitioned SSDU**         | 0.076   | 0.083   | 0.107   | 0.128   |
> | R=4              | A-DPS                           | 0.072   | 0.084   | 0.099   | 0.113   |
> | R=4              | **1D-Partitioned SSDU**         | 0.099   | 0.103   | 0.114   | 0.127   |
> | R=6              | A-DPS                           | 0.084   | 0.094   | 0.106   | 0.118   |
> | R=6              | **1D-Partitioned SSDU**         | 0.081   | 0.103   | 0.121   | 0.134   |
> | R=8              | A-DPS                           | 0.099   | 0.107   | 0.117   | 0.126   |
> | R=8              | **1D-Partitioned SSDU**         | 0.083   | 0.114   | 0.134   | 0.148   |
>
> | **Training Data** | **Reconstruction Method**  | **LPIPS (R=2)** | **DISTS (R=2)** | **LPIPS (R=4)** | **DISTS (R=4)** | **LPIPS (R=6)** | **DISTS (R=6)** | **LPIPS (R=8)** | **DISTS (R=8)** |
> |-------------------|----------------------------|------------------|------------------|------------------|------------------|------------------|------------------|------------------|------------------|
> | R=1              | FS-DPS                     | 0.061           | 0.028           | 0.106           | 0.059           | 0.166           | 0.094           | 0.231           | 0.126           |
> | R=2              | A-DPS                      | 0.118           | 0.062           | 0.127           | 0.068           | 0.138           | 0.076           | 0.150           | 0.082           |
> | R=2              | **1D-Partitioned SSDU**    | 0.141           | 0.084           | 0.144           | 0.081           | 0.187           | 0.108           | 0.223           | 0.129           |
> | R=4              | A-DPS                      | 0.155           | 0.090           | 0.155           | 0.092           | 0.161           | 0.095           | 0.168           | 0.099           |
> | R=4              | **1D-Partitioned SSDU**    | 0.169           | 0.105           | 0.169           | 0.092           | 0.187           | 0.104           | 0.214           | 0.123           |
> | R=6              | A-DPS                      | 0.180           | 0.110           | 0.178           | 0.110           | 0.181           | 0.111           | 0.186           | 0.114           |
> | R=6              | **1D-Partitioned SSDU**    | 0.143           | 0.092           | 0.169           | 0.102           | 0.196           | 0.117           | 0.222           | 0.133           |
> | R=8              | A-DPS                      | 0.212           | 0.129           | 0.204           | 0.127           | 0.204           | 0.127           | 0.204           | 0.127           |
> | R=8              | **1D-Partitioned SSDU**    | 0.142           | 0.094           | 0.176           | 0.109           | 0.206           | 0.127           | 0.231           | 0.140           |
>
> (cont')

---

> ### Author Response · Authors · 2024-11-20
> **Response to Reviewer Q5u6 (cont')**
>
> **Feedback**: I am confused by the bolding in Table 2. For methods trained with clean data, no values are bolded. For the R=2 and R=4 sections, A-DPS has equal performance to A-ALD in some metrics. The A-ALD metrics that are equally performant should be bolded for fairness. Please revise this table so that it is consistent.
>
> **Response**: We thank the Reviewer for pointing out the inconsistencies in Table 2. We have revised the table to ensure consistent bolding throughout. Initially, metrics for methods trained with clean data ($R=1$) were not bolded as our primary focus was on comparisons between self-supervised approaches. However, based on your feedback, we have now highlighted the best performance for each inference $R$ within each training scenario, including for methods trained with clean data. Additionally, the bolding issue with metrics for our proposed methods, A-ALD and A-DPS has been corrected to ensure fairness and consistency.
>
> ---
>
> **See weaknesses. Although, I do have one actual question:**
>
> **Question**: Did you consider training your MRI diffusion model (ambient or other) with the multi-coil data? To be clear: I am not asking you to do this, as it is a fundamental departure from what you have here. That said, I wonder if you tried this. I know that in, e.g., fastMRI, the number of coils is inconsistent, but you can get around this issue with virtual coils [1]. I suspect that the overall performance could be further improved with a multi-coil diffusion process, rather than a coil-combined one.
>
> [1] T. Zhang, J. M. Pauly, S. S. Vasanawala, and M. Lustig, “Coil compression for accelerated imaging with Cartesian sampling,” 2013
>
> **Response**: We thank the Reviewer for this insightful question. Currently, we define A_train according to Eq. 2.7. This is convenient because it lets us use an image-to-image network architecture such as that used by EDM. However, an alternative could be to use the A operator directly as in Eq. 2.5, in which case the network would go from multi-coil k-space to an image. This network could be preceded by an IFFT and would effectively be a multi-coil-to-image network, and could be standardized using coil compression [1]. We agree that such a network architecture could be interesting and potentially more expressive, as we currently collapse the multi-coil information through zero-filling adjoint. Another possibility is to define A_train as the pseudo-inverse applied to A. We appreciate the suggestion and view this as an interesting direction for future research. We have incorporated this discussion in the future directions section of our revised manuscript.

---

> > ### Comment · Reviewer_Q5u6 · 2024-11-21
> >
> > I thank the authors for their replies to my concerns and for answering my question.
> >
> > After reading the author replies and the other reviews, I will keep my current overall score of 8.

---

### Official Review · Reviewer_gieB · 2024-10-24

**Soundness:** 2
**Presentation:** 3
**Contribution:** 3
**Rating:** 8
**Confidence:** 4

**Summary:**

The paper proposes a posterior sampling method for inverse problems called Ambient Diffusion Posterior Sampling (A-DPS), which utilizes a diffusion model trained only on corrupted images. They modify the existing Ambient Diffusion framework to work for the accelerated MRI problem, allowing the training of an unconditional diffusion model with only subsampled MRI measurements. Then, they integrate the Ambient Diffusion models into the DPS framework to form a posterior sampling approach. The experiments on accelerated MRI and natural image datasets consistently show better recovery performance than the state-of-the-art DPS when the conditioning is highly corrupted but not when the corruption level is low.

**Strengths:**

- The paper addresses a significant problem in solving inverse problems when ground truth data is unavailable. By only requiring corrupted measurements for training, the method presents an avenue to solve a wider set of inverse problems.
- The framework is novel as it incorporates the benefits of unconditional Ambient Diffusion with the conditional flexibility of DPS.
- The experiments are adequately extensive and effectively demonstrate the benefits (and shortcomings) of their approach compared to several existing methods.

**Weaknesses:**

While the general methodology is explained well, the description of particular details could be modified to improve clarity and soundness
- Eq: 2.9: Are these equations correct? It seems to me that it should be $y_{t,\text{train}} = A_{\text{train}}(y_{\text{train}} + \sigma_{t}\eta)$ if $A_{\text{train}}$ is an orthogonal projection matrix or $y_{t,\text{train}} = A_{\text{train}}(x_{\text{train}} + \sigma_{t}\eta)$ if not
- What is the input to the diffusion model $h_\theta$? In Eq 2.10, it takes the additional subsampling matrix $\tilde{P}$. In Eq 2.11, it takes the full forward operator $A_\text{train}$.
- An intuitive explanation of $h_{\theta}(\tilde{y}\_{t, t_{\text{train}}}, \tilde{P})$ would be helpful for understanding the overall algorithm. Specifically, what does the training accomplish and why can this be directly plugged into Eq. 2.11?
- What is the reason or intuition for why an A-DPS trained on one corruption can generate posterior samples for a different corruption? A good hypothesis could better sell this property of the method.

Minor Points:
- L93-94: This sentence is confusing because it sounds like the Ambient Diffusion framework is "the first framework to solve inverse problems with diffusion models learned from linearly corrupted data". I would reword this to avoid confusion.
- Eq 2.4: It was not clear what $\tilde{A}_\text{train}$ represents. I would explain this matrix first before the equation, so the equation is more interpretable
- L255: Typo with "In that regards the multiplicity..."
- L364: Typo with "Similar to the results in The authors of ..."
- The construction of Fig 3 is confusing with the labeling of "Recon-FS". Since it's vertically displaced, it is easy to misread. I would suggest changing this to "Error" or rotate it vertically so "Recon-FS" is on a single line

**Questions:**

- Would it work better to train a single network on multiple acceleration rates rather than a separate network for each acceleration rate?
- Were the evaluations done on a single posterior sample for each method? Or do you generate many posterior samples and compute a posterior mean?

---

> ### Author Response · Authors · 2024-11-20
> **Response to Reviewer gieB**
>
> We thank a lot the Reviewer for their time and their valuable Review. We are glad that the Reviewer appreciated the novelty of our proposed algorithm and the extensiveness of our evaluation strategy.
>
> **Feedback**: Eq: 2.9: Are these equations correct?
>
> **Response**: The Reviewer is absolutely right, there was a typo in the Equations. We fixed the typo and we updated the Equations 2.9 and A.3 in our revised submission. We thank the Reviewer for careful reading!
>
> ---
>
> **Feedback**: What is the input to the diffusion model ? In Eq 2.10, it takes the additional subsampling matrix . In Eq 2.11, it takes the full forward operator
>
> **Response**: The models from the Ambient Diffusion paper that were trained for natural images, take the matrix tilde $A$ as input. The Ambient Diffusion MRI models we trained for this paper, take $\tilde{P}$ as input. We updated the notation to make this point explicit. We thank the Reviewer for their useful feedback.
>
> ---
>
> **Feedback**: An intuitive explanation of would be helpful for understanding the overall algorithm. Specifically, what does the training accomplish and why can this be directly plugged into Eq. 2.11?
>
> **Response**: The training objective of Ambient Diffusion leads to a model that estimates $E[x_0 | \tilde{A} x_t, \tilde{A}]$ (or in the case of the MRI models $E[x_0 | \tilde{P} x_t, \tilde{P}])$. This is used in equation 2.11 to obtain the one-step prediction for $x_0$ from the current corrupted iterate $x_t$. Specifically, given a current iterate $x_t$, we create the linear measurements $Ax_t$ and we pass them through the model that is trained to obtain an estimation of $x_0$ given linear measurements with noise. We added this clarification in our revised submission.
>
> ---
>
> **Feedback**: What is the reason or intuition for why an A-DPS trained on one corruption can generate posterior samples for a different corruption? A good hypothesis could better sell this property of the method.
>
> **Response**: The result of the Ambient Diffusion training process is a generative model. Generative models can be used at inference time to solve a variety of different inverse problems. This idea was proposed for GANs in the CSGM paper and has been extended to diffusion algorithms and other classes of generative models. Our trained models with Ambient Diffusion leverage a DPS-type algorithm (A-DPS) to guide the generations towards solutions that explain the measurements. That’s intuitively what’s going on in this paper.
>
> ---
>
> **Minor Points:**
>
> **Feedback**: L93-94: This sentence is confusing because it sounds like the Ambient Diffusion framework is "the first framework to solve inverse problems with diffusion models learned from linearly corrupted data". I would reword this to avoid confusion.
>
> **Response**: We rephrased this to: “We propose the first framework to solve inverse problems with Ambient Diffusion models. These are models that are trained using only access to linear measurements and they estimate the ambient score [...]”.
>
> ---
>
> **Feedback**: Eq 2.4: It was not clear what  represents. I would explain this matrix first before the equation, so the equation is more interpretable
>
> **Response**: Good point, we updated the submission to include a short description of $\tilde{A}$ before the equation.
>
> ---
>
> **Feedback**: L255: Typo with "In that regards the multiplicity..."
>
> **Response**: Fixed, we thank the Reviewer for careful reading!
>
> ---
>
> **Feedback**: L364: Typo with "Similar to the results in The authors of ..."
>
> **Response**: Fixed, we thank the Reviewer for careful reading!
>
> ---
>
> **Feedback**: The construction of Fig 3 is confusing with the labeling of "Recon-FS". Since it's vertically displaced, it is easy to misread. I would suggest changing this to "Error" or rotate it vertically so "Recon-FS" is on a single line
>
> **Response**: We have updated the label to "Error (10x)" to improve clarity and ensure it is easier to interpret.
>
> ---
>
> **Questions**
>
> **Question**: Would it work better to train a single network on multiple acceleration rates rather than a separate network for each acceleration rate?
>
> **Response**: Training a model at different corruption levels could lead to improved performance, e.g. this [paper](https://arxiv.org/abs/2411.02780) shows that a few good points can improve the performance of Ambient Diffusion models. For all our experiments, we assumed access to homogeneous data, but it is certainly interesting for future work to look at heterogeneous settings.
>
> ---
>
> **Question**: Were the evaluations done on a single posterior sample for each method? Or do you generate many posterior samples and compute a posterior mean?
>
> **Response**: Initially, we conducted a subset of experiments using multiple posteriors and observed that averaging posteriors led to slight improvements in metrics, but the benefits were consistent across all reconstruction algorithms. Due to computational and time constraints, we opted to evaluate single posterior samples.

---

> > ### Comment · Reviewer_gieB · 2024-11-21
> >
> > The authors have addressed most of my concerns, so I have subsequently raised my overall score to an 8 and the presentation to a 3. Overall, I think the paper addresses an important problem of solving linear inverse problems without access to ground truth data, and the results demonstrate this is a promising direction to address the problem.
> >
> > It is still not clear to me why an A-DPS model trained on say inpainting can be used for super-resolution at inference time as in Sec. 3.3. The authors claim that it is a result of the A-DPS model being a generative model and other works like CSGM demonstrate it can be used to solve a variety of inverse problems. However, CSGM and the original DPS are able to do this because they train an unconditional model, then use the condition to guide the generation at inference time. In Sec. 3.3, the authors train a conditional model, which takes the forward operator $A_{\text{train}}$, then use a different forward operator $A_{\text{inf}}$ at inference. It's not very intuitive to me that the network would be able to work with a condition it has never seen before.

---

> > > ### Author Response · Authors · 2024-11-21
> > > **Further clarification**
> > >
> > > Thank you for reading our rebuttal and for raising your score. We are very glad that we managed to address most of your concerns.
> > >
> > > Regarding your final question, let us clarify: our Ambient Model does not ever take A_inf as input (see Equation 2.11).
> > > Assume that you are at a point x_t and you want to move towards an x_0 that satisfies the measurements.
> > >
> > > DPS would move using guidance from two terms: the denoising term using the score and the measurements matching term.
> > > The denoising term essentially estimates the noise that needs to be subtracted and the measurements matching term is the gradient of the error in measurements space between the available measurements and the best prediction we have for x0 given xt (which is x0_hat, the conditional expectation).
> > >
> > > Ambient DPS does something similar. We also have two terms. For the denoising term, instead of the proper score, we have the Ambient Score. It serves essentially the same purpose: it tries to estimate the noise that needs to be subtracted. For the measurements matching term, we once again measure the error between the measurements and the most likely prediction that we have for x0 given xt. The main difference between Ambient DPS and DPS is that with Ambient DPS we only have a network that takes tilde A_{train}x_t as input, so for each current iterate x_t, we first have to move it in the distribution that the network is trained with (A_{train}x_t) and then use that input for our best prediction of x0.
> > >
> > > We hope that helps! Once again, thank you for your time and for your high-quality review.

---

### Official Review · Reviewer_GFMn · 2024-10-26

**Soundness:** 3
**Presentation:** 4
**Contribution:** 3
**Rating:** 8
**Confidence:** 5

**Summary:**

This paper proposes a novel method for learning a diffusion model from corrupted data, supported by a theoretical analysis of training the MMSE restoration estimator under such conditions. Empirical results indicate that the learned MMSE restoration estimator can operate similarly to an MMSE denoiser, enabling effective score-based generation and restoration tasks. Overall, this is a highly interesting paper with the potential to open many new research directions. However, certain aspects of the analysis and discussion could benefit from further improvement.

**Strengths:**

(1) This paper presents a novel framework for learning a ‘diffusion model’ solely from corrupted data, which is especially valuable in the field of medical imaging reconstruction, where only undersampled data is often available.
(2) A solid theoretical proof for learning MMSE restoration estimator from corrupted data under suitable assumption.
(3) Based on this diffusion model, the authors demonstrate its effectiveness in both generative tasks and solving inverse problems, highlighting the efficiency and potential impact of this approach.

**Weaknesses:**

(1) This work introduces an intriguing approach to learning a diffusion model from corrupted data. However, the authors’ analysis assumes that the learned MMSE restoration estimator can approximate a learned MMSE denoising estimator to derive a score function for sampling—an assumption that could benefit from further clarification. In approaches like DPS, posterior sampling depends on the score derived from Tweedie’s formula, which is specifically suited for denoisers. In this study, however, the corresponding score function is not discussed in sufficient detail. A discussion of recent approaches such as DRP[1] and SHARP[2], which utilize the MMSE restoration estimator with a modified score function, could enhance the analysis and provide a more fitting framework for this model.

(2) The theoretical approach in this work is compelling and well-suited for training with randomly inpainted images. However, in the MRI setting used here, different training samples may share the auto-calibration signal (ACS) region, potentially leading to an over-representation of corresponding k-space regions in the training data. A more solid approach might involve introducing a diagonal weighting matrix in the loss function to account for these oversampled regions. A similar work discussing this point for MRI reconstruction is [3].

(3) Further explanation to how the FID value is calculated for MRI data will be helpful.

(4) Some minor suggestions regarding the paper's presentation: (1) In Table 3, it would be beneficial to standardize the significant digits for the FID values to improve readability. (2) For Figures 7 and 8, adjusting the color bar could enhance visual clarity and make the figures easier to interpret.

Reference:
[1] Y. Hu, M. Delbracio, P. Milanfar, and U. S. Kamilov, “A Restoration Network as an Implicit Prior,” Proc. Int. Conf. Learn. Represent. (ICLR 2024) (Vienna, Austria, May 7-11).
[2] Y. Hu, A. Peng, W. Gan, P. Milanfar, M. Delbracio, and U. S. Kamilov, “Stochastic Deep Restoration Priors for Imaging Inverse Problems.”  [https://arxiv.org/abs/2410.02057]
[3] C. Millard and M. Chiew, "A Theoretical Framework for Self-Supervised MR Image Reconstruction Using Sub-Sampling via Variable Density Noisier2Noise," in IEEE Transactions on Computational Imaging, vol. 9, pp. 707-720,

**Questions:**

The questions have been covered in the weakness part.

---

> ### Author Response · Authors · 2024-11-20
> **Response to Reviewer GFMn**
>
> We thank a lot the Reviewer for their time and their valuable Review. We are glad that the Reviewer appreciated the novelty of our proposed algorithm and the overall presentation of our findings.
>
> **Feedback**: The theoretical approach in this work is compelling and well-suited for training with randomly inpainted images. However, in the MRI setting used here, different training samples may share the auto-calibration signal (ACS) region, potentially leading to an over-representation of corresponding k-space regions in the training data. A more solid approach might involve introducing a diagonal weighting matrix in the loss function to account for these oversampled regions. A similar work discussing this point for MRI reconstruction is [3].
>
> **Response**: The auto-calibration signal (ACS) region is indeed shared across training samples, which could lead to an over-representation of certain k-space regions. While our current approach does not explicitly address this, we agree that introducing a diagonal weighting matrix in the loss function, as suggested, is a promising direction to mitigate this issue. We have included a discussion regarding this and reference [3] in the limitations and future directions section of our manuscript.
>
> Furthermore, we have implemented the self-supervised baseline in [3] and have included these results under Table 2 (NRMSE, SSIM, PSNR) and Table 3 (LPIPS, DISTS). Similar to our previous results, we show that A-DPS outperforms other algorithms at high acceleration rates.
>
> [3] C. Millard and M. Chiew, "A Theoretical Framework for Self-Supervised MR Image Reconstruction Using Sub-Sampling via Variable Density Noisier2Noise," in IEEE Transactions on Computational Imaging
>
> Below we report NRMSE, LPIPS, and DISTS across 100 validation samples:
>
> | **Training Data** | **Reconstruction Method**       | **R=2** | **R=4** | **R=6** | **R=8** |
> |-------------------|---------------------------------|---------|---------|---------|---------|
> | R=1              | FS-DPS                          | 0.034   | 0.065   | 0.121   | 0.179   |
> | R=2              | A-DPS                           | 0.058   | 0.074   | 0.093   | 0.112   |
> | R=2              | **1D-Partitioned SSDU**         | 0.076   | 0.083   | 0.107   | 0.128   |
> | R=4              | A-DPS                           | 0.072   | 0.084   | 0.099   | 0.113   |
> | R=4              | **1D-Partitioned SSDU**         | 0.099   | 0.103   | 0.114   | 0.127   |
> | R=6              | A-DPS                           | 0.084   | 0.094   | 0.106   | 0.118   |
> | R=6              | **1D-Partitioned SSDU**         | 0.081   | 0.103   | 0.121   | 0.134   |
> | R=8              | A-DPS                           | 0.099   | 0.107   | 0.117   | 0.126   |
> | R=8              | **1D-Partitioned SSDU**         | 0.083   | 0.114   | 0.134   | 0.148   |
>
> | **Training Data** | **Reconstruction Method**  | **LPIPS (R=2)** | **DISTS (R=2)** | **LPIPS (R=4)** | **DISTS (R=4)** | **LPIPS (R=6)** | **DISTS (R=6)** | **LPIPS (R=8)** | **DISTS (R=8)** |
> |-------------------|----------------------------|------------------|------------------|------------------|------------------|------------------|------------------|------------------|------------------|
> | R=1              | FS-DPS                     | 0.061           | 0.028           | 0.106           | 0.059           | 0.166           | 0.094           | 0.231           | 0.126           |
> | R=2              | A-DPS                      | 0.118           | 0.062           | 0.127           | 0.068           | 0.138           | 0.076           | 0.150           | 0.082           |
> | R=2              | **1D-Partitioned SSDU**    | 0.141           | 0.084           | 0.144           | 0.081           | 0.187           | 0.108           | 0.223           | 0.129           |
> | R=4              | A-DPS                      | 0.155           | 0.090           | 0.155           | 0.092           | 0.161           | 0.095           | 0.168           | 0.099           |
> | R=4              | **1D-Partitioned SSDU**    | 0.169           | 0.105           | 0.169           | 0.092           | 0.187           | 0.104           | 0.214           | 0.123           |
> | R=6              | A-DPS                      | 0.180           | 0.110           | 0.178           | 0.110           | 0.181           | 0.111           | 0.186           | 0.114           |
> | R=6              | **1D-Partitioned SSDU**    | 0.143           | 0.092           | 0.169           | 0.102           | 0.196           | 0.117           | 0.222           | 0.133           |
> | R=8              | A-DPS                      | 0.212           | 0.129           | 0.204           | 0.127           | 0.204           | 0.127           | 0.204           | 0.127           |
> | R=8              | **1D-Partitioned SSDU**    | 0.142           | 0.094           | 0.176           | 0.109           | 0.206           | 0.127           | 0.231           | 0.140           |
>
> (cont')

---

> ### Author Response · Authors · 2024-11-20
> **Response to Reviewer GFMn (cont')**
>
> **Feedback**: Further explanation to how the FID value is calculated for MRI data will be helpful.
>
> **Response**: To quantify unconditional sample quality, we calculate FID scores using a pre-trained VGG network, following the methodology outlined in [4]. Specifically for MRI data, the k-space reconstructions are converted to magnitude images, and the FID is computed using embeddings from the convolutional layers of a VGG network trained on natural images. While VGG is not optimized for MRI data, it provides a perceptual metric that aligns with image sharpness and structural consistency, as validated in related works like [A].
>
> We also compute the distribution metrics (Conditional FID) between samples from the posterior and the ground-truth distribution, using the same VGG network (included in Table 1). We have clarified this process in the revised manuscript.
>
> [A] Bendel, Matthew, Rizwan Ahmad, and Philip Schniter. "A regularized conditional GAN for posterior sampling in image recovery problems." Advances in Neural Information Processing Systems 36 (2024).
>
> ---
>
> **Feedback**: Some minor suggestions regarding the paper's presentation: (1) In Table 3, it would be beneficial to standardize the significant digits for the FID values to improve readability. (2) For Figures 7 and 8, adjusting the color bar could enhance visual clarity and make the figures easier to interpret.
>
> **Response**:
> 1. We thank the Reviewer for their suggestion regarding Table 3. We have combined Tables 1 and 3 into a single, standardized Table 1 in the main paper. This table now presents both Unconditional and Posterior Conditional FID scores with all values standardized for improved readability. Additionally, we have highlighted the best-performing method for each training acceleration factor $R$ to enhance visibility.
>
> 2. Based on the Reviewer’s feedback, we have removed Figures 7 and 8. In their place, we have introduced an expanded Table 2, which extends the posterior reconstruction metrics table (Table 2) by including SSIM and PSNR metrics alongside NRMSE. We have also included a new Table 3 where we report LPIPS and DISTS for the accelerated MR reconstruction experiments. Significant values are highlighted in these tables to improve clarity and emphasize key results.

---

> ### Comment · Reviewer_GFMn · 2024-11-21
> **Response to authors**
>
> Thank you to the authors for thoughtfully addressing the points raised in my review. After carefully reviewing the responses, I find that most of my concerns have been adequately resolved. As a result, I will maintain my overall score (6: marginally above the acceptance threshold) while increasing the soundness rating to 3 and presentation rating to 4.

---

> > ### Author Response · Authors · 2024-11-21
> > **Thank you!**
> >
> > Thank you for going over our rebuttal.
> > Please let us know if there is anything we could do to make you consider increasing further your rating.
> >
> > In any case, thank you; your review helped us improve our work, which is the most important outcome of the reviewing process.

---

> ### Comment · Reviewer_GFMn · 2024-11-21
> **Response to authors**
>
> Thank you! Overall, I find this to be a strong paper, particularly in the context of learning a diffusion model from undersampled MRI data, which has significant potential for practical applications. However, there are two key reasons why I did not raise my score to 8.
>
> 1. Is there a more formal mathematical explanation for why DM learned from undersampled or corrupted data can get better performance in solving certain inverse problems?
>
> 2. For MRI applications, validating the algorithm on real-world scenarios where fully sampled data is unattainable (e.g., dynamic liver MRI) would further strengthen the paper's impact and practical relevance.

---

> > ### Author Response · Authors · 2024-11-21
> > **Response to Reviewer GFMn**
> >
> > We thank the reviewer for engaging with us. We agree that including experiments on prospectively accelerated MRI data where no ground truth are available would strengthen the impact of our work. We will make our best effort to complete this task before the end of the rebuttal period.
> >
> > We would like to point out that our work is a proof of principle of the approach, much like References [2] and [3] mentioned by the Reviewer, which also did not apply their method to prospectively accelerated MRI data

---

> > > ### Author Response · Authors · 2024-11-30
> > > **Additional experiments**
> > >
> > > Dear Reviewer,
> > > following your recommendation, we worked hard to obtain results for MRI data where no ground truth is available.
> > >
> > > Specifically, we downloaded real data from the following link: https://old.mridata.org/undersampled/abdomens and we trained a model using Ambient Diffusion.
> > >
> > > In the following image you can view training samples and generated samples using our trained model: https://ibb.co/HPwkCY8
> > >
> > > Further, we used Ambient DPS to perform posterior sampling using the given data for which no ground truth is available. The results we get are included in the following link: https://ibb.co/7SDzWzZ
> > >
> > > Our results are still very preliminary because of the short time we had available for our rebuttal, e.g. we are noticing that the models performance is still increasing as we train for longer. We will work on improving these results and including a detailed comparison with the baselines for the camera-ready version o the paper.
> > >
> > > Finally, regarding your first question, "Is there a more formal mathematical explanation for why DM learned from undersampled or corrupted data can get better performance in solving certain inverse problems?": we are not able to provide a formal proof since the performance of DPS itself is not easy to characterize beyond very simple distributions, but we believe that the phenomenon is related to a bias-variance tradeoff. Our current hypothesis is that the corruption in the data acts as regularization and that causes the learned prior to generalize better to novel corruptions at inference time.

---

> ### Comment · Reviewer_GFMn · 2024-11-30
> **Response to authors**
>
> Thank you for providing the additional experiments on MRI data where no ground truth is available. These experiments addressed most of my concerns, so I am increasing my score to 8. I also appreciate the authors' efforts to make the paper more comprehensive overall.
>
> Regarding the question: "Is there a more formal mathematical explanation for why diffusion models learned from undersampled or corrupted data can achieve better performance in solving certain inverse problems?"—I generally agree with your intuition. To the regularization you mentioned, I suggest considering the perspective I outlined in the first round (which is similar to but distinct from DRP [1]). I suggest including a discussion or future work section to elaborate on your hypothesis.
>
> [1] Y. Hu, M. Delbracio, P. Milanfar, and U. S. Kamilov, “A Restoration Network as an Implicit Prior,” Proc. Int. Conf. Learn. Represent. (ICLR 2024) (Vienna, Austria, May 7-11).

---

### Official Review · Reviewer_GxHM · 2024-11-04

**Soundness:** 3
**Presentation:** 3
**Contribution:** 2
**Rating:** 5
**Confidence:** 5

**Summary:**

This paper applies the recent advancement in self-supervised learning for diffusion models, named ambient diffusion, to inverse problems. The effectiveness of this approach is shown through its application to MRI reconstruction and image inpainting.

**Strengths:**

1) The paper represents an early investigation into the application of self-supervised learning for diffusion models in the context of inverse problems.
2) The paper is well-organized, and the presentation is easy to follow.

**Weaknesses:**

1) Although this paper introduces the first self-supervised diffusion method for inverse problems, the overall novelty is somewhat limited. It appears to be a straightforward integration of existing ambient diffusion with well-known diffusion posterior sampling, with application to MRI reconstruction. While the investigation holds value from a research perspective, the new insight/concept might be limited.

2) In the training phase (as described in Equation 2.10), the network takes \tilde{A} as input, considering one type of imaging artifact. However, during testing (Equation 2.11), the network uses A as input, which features a different imaging artifact. This inconsistency raises concerns about potential suboptimal performance. Could the authors clarify this aspect?

3) There is insufficient discussion regarding why the proposed A-DPS method outperforms FS-DPS, especially at high undersampling rates. Given that FS-DPS uses a diffusion model trained on fully sampled data, the results seem unexpectedly favorable.

4) The manuscript lacks discussions with recent theoretical developments in self-supervised learning for MRI [1,2], which share similar proof strategy with Theorem 1 in this paper.

5) The numerical validation could be improved, with certain experimental setups are not fully explored (see specific questions below). Additionally, the baseline methods for MRI reconstruction cited are well-known but somewhat dated (2019, 2020). Given the extensive literature on MRI reconstruction, these baselines may not represent the current state of the field. Comparing recent alternatives, such as [1-3] and self-supervised or supervised methods in survey [3] and [4], respectively, could improve this study. There is also no comparison between A-OS and other end-to-end methods. Despite that, I fully understand the time constraints of the rebuttal period and do not intend to suggest that the authors include comparisons with new baselines.

I am willing to increase my score if the authors could address my concerns.

[1] Millard, Charles, and Mark Chiew. "A theoretical framework for self-supervised MR image reconstruction using sub-sampling via variable density Noisier2Noise." IEEE transactions on computational imaging (2023).

[2] Gan, Weijie, et al. "Self-supervised deep equilibrium models with theoretical guarantees and applications to MRI reconstruction." IEEE Transactions on Computational Imaging (2023).

[3] Sriram, Anuroop, et al. "End-to-end variational networks for accelerated MRI reconstruction." Medical Image Computing and Computer Assisted Intervention–MICCAI 2020: 23rd International Conference, Lima, Peru, October 4–8, 2020, Proceedings, Part II 23. Springer International Publishing, 2020.

[4] Akçakaya, Mehmet, et al. "Unsupervised deep learning methods for biological image reconstruction and enhancement: An overview from a signal processing perspective." IEEE Signal Processing Magazine 39.2 (2022): 28-44.

[5] Heckel, Reinhard, et al. "Deep learning for accelerated and robust MRI reconstruction: a review." arXiv preprint arXiv:2404.15692 (2024).

**Questions:**

1) The further corruption ratio is simply R+1. How does this further corruption ratio impact the performance, says R+2, R+3, etc?
2) Does the supervised DPS share the same network architecture with the proposed method?
3) How does the coil sensitivity map in this training dataset and the L1-wavelet baseline be estimated?
4) What is the time complexity of the methods?
5) What is the actual under sampled ratio considering the sampling mask for different R has the same 20 lines of the auto calibration signal?

---

> ### Author Response · Authors · 2024-11-20
> **Response to Reviewer GxHM**
>
> We thank a lot the Reviewer for their time and their valuable Review. We are glad that the Reviewer appreciated the effectiveness of our proposed algorithm and the presentation of our work.
>
> ---
>
> **Feedback**: Although this paper introduces the first self-supervised diffusion method for inverse problems, the overall novelty is somewhat limited. It appears to be a straightforward integration of existing ambient diffusion with well-known diffusion posterior sampling, with application to MRI reconstruction. While the investigation holds value from a research perspective, the new insight/concept might be limited.
>
> **Response**: While we agree that extending DPS to A-DPS is technically straightforward, we believe the novelty lies in the finding that models trained on corrupted data can outperform those trained on clean data, especially for severely ill-posed problems. Furthermore, we draw a parallel to the DPS algorithm, which uses a simple approximation to address the intractability of the likelihood term. Despite its simplicity, DPS has been widely adopted and demonstrated significant success across diverse applications. Similarly, while A-DPS represents an adaptation of DPS within the Ambient Diffusion framework, the experimental results showcasing its efficacy in solving inverse problems highlight its importance and utility. We hope the reviewer recognizes this contribution as a valuable advancement for practical inverse problem-solving.
>
> ---
>
> **Feedback**: In the training phase (as described in Equation 2.10), the network takes \tilde{A} as input, considering one type of imaging artifact. However, during testing (Equation 2.11), the network uses A as input, which features a different imaging artifact. This inconsistency raises concerns about potential suboptimal performance. Could the authors clarify this aspect?
>
> **Response**: We thank the Reviewer for raising this important point. We noticed experimentally that the model is robust at inference time to matrices $A$, even when trained on matrices $\tilde{A}$. This is consistent with our experiments in Section 3.3 on pre-trained models for natural images. That said, in our codebase, we used $\tilde{A}$ for our experiments, so we updated Equation 2.11 to reflect this.
>
> ---
>
> **Feedback**: There is insufficient discussion regarding why the proposed A-DPS method outperforms FS-DPS, especially at high undersampling rates. Given that FS-DPS uses a diffusion model trained on fully sampled data, the results seem unexpectedly favorable.
>
> **Response**: Respectfully, we believe this observation is not a weakness but rather the main finding of our paper. The superior performance of A-DPS at high undersampling rates arises because models trained on corrupted data can be better suited to handle the ill-posed nature of inverse problems under severe corruption. By being exposed to corrupted training data, A-DPS can learn more robust priors that generalize better to high-acceleration regimes, unlike FS-DPS, which may overfit clean data and fail to adapt effectively to severe undersampling. We have validated this extensively across different data modalities and inverse problems in our experiments. Based on your feedback, we have incorporated this discussion in our revised submission.
>
> ---
>
> **Feedback**: The manuscript lacks discussions with recent theoretical developments in self-supervised learning for MRI [1,2], which share similar proof strategy with Theorem 1 in this paper.```
>
> **Response**: We thank the Reviewer for raising this point. We have included a discussion of these methods in our revised submission. Regarding [1], while it shares a similar proof strategy to Theorem 1 in our work, its primary focus is on learning a restoration model tailored to a specific corruption type. In comparison, Ambient Diffusion aims to learn a generative model, which provides greater flexibility at inference time to address a wider variety of corruption types and inverse problems. We believe this distinction highlights the broader applicability of our framework.
>
>
> (cont')

---

> ### Author Response · Authors · 2024-11-20
> **Response to Reviewer GxHM (cont')**
>
> **Feedback**: The numerical validation could be improved, with certain experimental setups are not fully explored (see specific questions below). Additionally, the baseline methods for MRI reconstruction cited are well-known but somewhat dated (2019, 2020). Given the extensive literature on MRI reconstruction, these baselines may not represent the current state of the field. Comparing recent alternatives, such as [1-3] and self-supervised or supervised methods in survey [3] and [4], respectively, could improve this study. There is also no comparison between A-OS and other end-to-end methods. Despite that, I fully understand the time constraints of the rebuttal period and do not intend to suggest that the authors include comparisons with new baselines.
>
> **Response**: We thank the Reviewer for these valuable suggestions. Our primary focus with MRI evaluation was on self-supervised baselines, as these align more closely with our core contributions. We believe that state-of-the-art FS-DPS provides the best fully supervised comparison. Additionally, we include end-to-end MoDL, which, while slightly dated, serves as a close proxy for [3].
>
> In regards to self-supervised baselines, we agree that including more recent baselines could enhance the study. We have implemented [1] and have included it in our revised submission under Table 2 (NRMSE, SSIM, PSNR) and Table 3 (LPIPS, DISTS). Similar to our previous results, we show that A-DPS outperforms other algorithms at high acceleration rates.
>
> [1] Millard, Charles, and Mark Chiew. "A theoretical framework for self-supervised MR image reconstruction using sub-sampling via variable density Noisier2Noise." IEEE transactions on computational imaging (2023).
>
> Below we report NRMSE, LPIPS, and DISTS across 100 validation samples:
>
> | **Training Data** | **Reconstruction Method**       | **R=2** | **R=4** | **R=6** | **R=8** |
> |-------------------|---------------------------------|---------|---------|---------|---------|
> | R=1              | FS-DPS                          | 0.034   | 0.065   | 0.121   | 0.179   |
> | R=2              | A-DPS                           | 0.058   | 0.074   | 0.093   | 0.112   |
> | R=2              | **1D-Partitioned SSDU**         | 0.076   | 0.083   | 0.107   | 0.128   |
> | R=4              | A-DPS                           | 0.072   | 0.084   | 0.099   | 0.113   |
> | R=4              | **1D-Partitioned SSDU**         | 0.099   | 0.103   | 0.114   | 0.127   |
> | R=6              | A-DPS                           | 0.084   | 0.094   | 0.106   | 0.118   |
> | R=6              | **1D-Partitioned SSDU**         | 0.081   | 0.103   | 0.121   | 0.134   |
> | R=8              | A-DPS                           | 0.099   | 0.107   | 0.117   | 0.126   |
> | R=8              | **1D-Partitioned SSDU**         | 0.083   | 0.114   | 0.134   | 0.148   |
>
> | **Training Data** | **Reconstruction Method**  | **LPIPS (R=2)** | **DISTS (R=2)** | **LPIPS (R=4)** | **DISTS (R=4)** | **LPIPS (R=6)** | **DISTS (R=6)** | **LPIPS (R=8)** | **DISTS (R=8)** |
> |-------------------|----------------------------|------------------|------------------|------------------|------------------|------------------|------------------|------------------|------------------|
> | R=1              | FS-DPS                     | 0.061           | 0.028           | 0.106           | 0.059           | 0.166           | 0.094           | 0.231           | 0.126           |
> | R=2              | A-DPS                      | 0.118           | 0.062           | 0.127           | 0.068           | 0.138           | 0.076           | 0.150           | 0.082           |
> | R=2              | **1D-Partitioned SSDU**    | 0.141           | 0.084           | 0.144           | 0.081           | 0.187           | 0.108           | 0.223           | 0.129           |
> | R=4              | A-DPS                      | 0.155           | 0.090           | 0.155           | 0.092           | 0.161           | 0.095           | 0.168           | 0.099           |
> | R=4              | **1D-Partitioned SSDU**    | 0.169           | 0.105           | 0.169           | 0.092           | 0.187           | 0.104           | 0.214           | 0.123           |
> | R=6              | A-DPS                      | 0.180           | 0.110           | 0.178           | 0.110           | 0.181           | 0.111           | 0.186           | 0.114           |
> | R=6              | **1D-Partitioned SSDU**    | 0.143           | 0.092           | 0.169           | 0.102           | 0.196           | 0.117           | 0.222           | 0.133           |
> | R=8              | A-DPS                      | 0.212           | 0.129           | 0.204           | 0.127           | 0.204           | 0.127           | 0.204           | 0.127           |
> | R=8              | **1D-Partitioned SSDU**    | 0.142           | 0.094           | 0.176           | 0.109           | 0.206           | 0.127           | 0.231           | 0.140           |
>
> (cont')

---

> ### Author Response · Authors · 2024-11-20
> **Response to Reviewer GxHM (cont')**
>
> **Question**: The further corruption ratio is simply R+1. How does this further corruption ratio impact the performance, says R+2, R+3, etc?
>
> **Response**: The further corruption during training should be as minimal as possible to avoid throwing away training data. We have not studied the effect of increased additional corruption at training time, but we believe that this would lead to performance deterioration.
>
> ---
>
> **Question**: Does the supervised DPS share the same network architecture with the proposed method?
>
> **Response**: Both models are based on the EDM codebase. The EDM model trained on the fully-sampled (R=1) dataset, as well as the four EDM models trained on L1-Wavelet reconstructions at each acceleration factor R  = 2, 4, 6, 8, were all trained with 65 million parameters. The Ambient Diffusion models on the other hand were trained with 36 million parameters, for faster training. There were no other differences in the network architecture.
>
> ---
>
> **Question**: How does the coil sensitivity map in this training dataset and the L1-wavelet baseline be estimated?
>
> **Response**: We estimate the sensitivity maps using ESPIRiT [A] from a fully sampled auto calibration signal (ACS).
>
> [A] Uecker, Martin, Peng Lai, Mark J. Murphy, Patrick Virtue, Michael Elad, John M. Pauly, Shreyas S. Vasanawala, and Michael Lustig. "ESPIRiT—an eigenvalue approach to autocalibrating parallel MRI: where SENSE meets GRAPPA." Magnetic resonance in medicine 71, no. 3 (2014): 990-1001.
>
> ---
>
> **Question**: What is the time complexity of the methods?
>
> **Response**: A-OS requires 1 function evaluation. The number of function evaluations required by A-DPS is the same as DPS and depends on the discretization used in the sampler.
>
> ---
>
> **Question**: What is the actual under sampled ratio considering the sampling mask for different R has the same 20 lines of the auto calibration signal?
>
> **Response**: The acceleration factor accounted for sampling of the central lines, i.e. it was defined as the matrix size divided by the number of sampled points.

---

> > ### Author Response · Authors · 2024-11-24
> > **Follow-up**
> >
> > Dear Reviewer, as the discussion period approaches its end, we would like to follow up and check if our rebuttal adequately addressed your questions.

---

> > > ### Comment · Reviewer_GxHM · 2024-11-25
> > >
> > > Thank you to the authors for their detailed response, which addresses some of my concerns. However, several issues remain unresolved, particularly regarding the novelty. As such, I will maintain my original score and also lean to weak rejection. Please see my comments below.
> > >
> > > (Weakness 1)
> > > The authors response that the main contribution of the paper is advancing practical inverse problem-solving. However, I find the practical impact of this work to be limited for the following reasons:
> > > (1) Both experiments are conducted on simulated data.
> > > (2) The Gaussian compressive sensing and super-resolution tasks, based on perfectly known models, are better characterized as proof-of-concept studies rather than practical, real-world problems.
> > > (3) More importantly, this paper dedicates significant space to MRI reconstruction, but practical impact in medical imaging typically requires real data acquisition and evidence of clinical relevance. It is uncommon to claim practical significance in medical imaging based on simulated data. There are specialized journals and conferences, such as Radiology and Magnetic Resonance in Medicine, that focus on MRI-specific problem-solving and might be a better fit for this work if its aim is practical impact in the medical imaging domain.
> > >
> > > Therefore, the response regarding the work's novelty is not convincing. While the combination of ambient diffusion and DPS is interesting, the overall contribution remains limited in scope.
> > >
> > > (Weakness 4)
> > > The authors suggest that prior works focus on restoration models, whereas this method employs a diffusion model. However, for a specific timestep t, the network is essentially trained to perform restoration/reconstruction. Thus, I find this distinction unconvincing, and my original concern remains.
> > >
> > > (Question 1)
> > > It remains unclear how the further corruption ratio relates to throwing away training data, as all available data are still used during training. Shouldn't this ratio only affect the way the network input is derived?
> > >
> > > (Questions 4 & 5)
> > > My concern is indeed about the actual value of the metric, not its formulation.

---

> > > > ### Author Response · Authors · 2024-11-30
> > > > **Further clarification**
> > > >
> > > > Dear Reviewer,
> > > >
> > > > we thank you for reading our rebuttal and for providing additional input. We are very glad that we were able to address some of your concerns. You are the only Reviewer currently leaning towards weak rejection and we will do our best to resolve any remaining issues.
> > > >
> > > > Regarding your first concern, the main argument seems to be that the practical impact of this work is limited because we are only experimenting with simulated corruptions where the forward model is known. We have two counter points (one of them includes **additional experiments**):
> > > >
> > > > 1) The vast majority of works on this topic also work with simulated data. The DPS paper itself (and the hundreds of papers on diffusion models for inverse problems that followed it) experiment and benchmark on simulated corruptions. In the MRI space itself, prior works on exactly this topic also work with simulated data: e.g. see: Y. Hu, A. Peng, W. Gan, P. Milanfar, M. Delbracio, and U. S. Kamilov, “Stochastic Deep Restoration Priors for Imaging Inverse Problems.” [https://arxiv.org/abs/2410.02057] and C. Millard and M. Chiew, "A Theoretical Framework for Self-Supervised MR Image Reconstruction Using Sub-Sampling via Variable Density Noisier2Noise.
> > > >
> > > > 2) Despite our work being a proof of concept, we worked hard during the rebuttal to obtain results on realistic MRI data where no ground-truth is available. Specifically, we trained an Ambient Diffusion model on data from here:  https://old.mridata.org/undersampled/abdomens . You can find training data and generated data at the following link: https://ibb.co/HPwkCY8 and also posterior samples in the following link: https://ibb.co/7SDzWzZ
> > > > These results are still very preliminary (e.g. performance improves as we keep training), but we wanted to demonstrate that our method could be potentially applied to these more realistic use-cases.
> > > >
> > > > Your next concern is that the difference between our generative model and restoration models is not very clear, since for each fixed $t$ we essentially learn a restoration model. Our answer is that there are two things that restoration models cannot do and our models can: 1) restoration models cannot generate new samples without some given corrupted input and 2) restoration models only work for the corruption they are trained. Our model can be used to solve arbitrary inverse problems at inference time. We refer the Reviewer to the CSGM paper for more details on generative priors for solving inverse problems, if this is of interest.
> > > >
> > > > Regarding the throwing away of data, our point is that if the model is trained with A_hat then it has to see A_hat during inference time as well. So we have to project each current iterate x_t with A_hat prior to feeding this to the model and this is harmful since we are throwing away part of our denoised input at each prediction step.

---

### Official Review · Reviewer_323G · 2024-11-04

**Soundness:** 2
**Presentation:** 4
**Contribution:** 3
**Rating:** 6
**Confidence:** 4

**Summary:**

This paper uses ambient diffusion models as a prior for solving ill-posed inverse problems. The ambient diffusion is trained on corrupted data acquired from a linear ill-posed operator. The authors also extend the ambient diffusions to use MRI k-space subsampled measurements as training data. The authors show the proposed framework can outperform DPS trained on clean images when dealing with severely ill-posed inverse problems.

**Strengths:**

1. The paper is motivated by the challenge of solving inverse problems without access to clean images, a common problem in real-world imaging applications.
2. Experimental results show the proposed framework is effective for severely ill-posed problems, sometimes outperforming priors trained on clean images.

**Weaknesses:**

1. Despite promising experiments, the novelty of the paper seems limited; the authors primarily extend ambient diffusion to MRI and adapt it within the DPS framework for solving inverse problems.

2. Below Eq. 2.11, the authors state, "We remark that similar to DPS, the proposed algorithm is an approximation to sampling from the true posterior distribution". However, the accuracy of this approximation should be rigorously compared to that of the original DPS, as it relies on estimating the score function from corrupted data.

3. The authors should consider including additional, recent self-supervised learning approaches as baselines, such as the following paper:
Gan, Weijie, et al. "Self-supervised deep equilibrium models with theoretical guarantees and applications to MRI reconstruction." IEEE Transactions on Computational Imaging (2023).

4. The paper needs to be further polished, for example, Eq 2.9 seems incorrect.

**Questions:**

Please take a look at the comments under weaknesses.

---

> ### Author Response · Authors · 2024-11-20
> **Response to Reviewer 323G**
>
> We thank a lot the Reviewer for their time and their valuable Review. We are glad that the Reviewer appreciated the effectiveness of our proposed algorithm.
>
> ---
>
> **Feedback**: Despite promising experiments, the novelty of the paper seems limited; the authors primarily extend ambient diffusion to MRI and adapt it within the DPS framework for solving inverse problems.
>
> **Response**:
> Thank you for the feedback. While we agree that extending DPS to A-DPS is technically straightforward, we believe the novelty lies in the finding that models trained on corrupted data can outperform those trained on clean data, especially for severely ill-posed problems. Furthermore, we draw a parallel to the DPS algorithm, which uses a simple approximation to address the intractability of the likelihood term. Despite its simplicity, DPS has been widely adopted and demonstrated significant success across diverse applications. Similarly, while A-DPS represents an adaptation of DPS within the Ambient Diffusion framework, the experimental results showcasing its efficacy in solving inverse problems highlight its importance and utility. We hope the reviewer recognizes this contribution as a valuable advancement for practical inverse problem-solving.
>
> ---
>
> **Feedback**: Below Eq. 2.11, the authors state, "We remark that similar to DPS, the proposed algorithm is an approximation to sampling from the true posterior distribution". However, the accuracy of this approximation should be rigorously compared to that of the original DPS, as it relies on estimating the score function from corrupted data.
>
> **Response**: The accuracy of the DPS approximation itself is poorly understood – the authors of DPS only provide a rough characterization of the Jensen gap in Theorem 1 of their paper. Following the Reviewer’s recommendation, we included the same result for A-DPS in our paper in Equation 2.12 of the updated submission.
>
> ---
>
> **Feedback**: The authors should consider including additional baselines [...]
>
> **Response**:
> We thank the Reviewer for their suggestion. We implemented another recent self-supervised baseline stemming from the SSDU framework [1]. We have included these results in our revised submission under Table 2 (NRMSE, SSIM, PSNR) and Table 3 (LPIPS, DISTS). Similar to our previous results, we show that A-DPS outperforms other algorithms at high acceleration rates.
>
> [1] Millard, Charles, and Mark Chiew. "A theoretical framework for self-supervised MR image reconstruction using sub-sampling via variable density Noisier2Noise." IEEE Transactions on Computational Imaging (2023).
>
> Below we report Normalized Root Mean Squared Error (NRMSE) across 100 validation samples:
>
> | **Training Data** | **Reconstruction Method**       | **R=2** | **R=4** | **R=6** | **R=8** |
> |-------------------|---------------------------------|---------|---------|---------|---------|
> | R=1              | FS-DPS                          | 0.034   | 0.065   | 0.121   | 0.179   |
> | R=2              | A-DPS                           | 0.058   | 0.074   | 0.093   | 0.112   |
> | R=2              | **1D-Partitioned SSDU**         | 0.076   | 0.083   | 0.107   | 0.128   |
> | R=4              | A-DPS                           | 0.072   | 0.084   | 0.099   | 0.113   |
> | R=4              | **1D-Partitioned SSDU**         | 0.099   | 0.103   | 0.114   | 0.127   |
> | R=6              | A-DPS                           | 0.084   | 0.094   | 0.106   | 0.118   |
> | R=6              | **1D-Partitioned SSDU**         | 0.081   | 0.103   | 0.121   | 0.134   |
> | R=8              | A-DPS                           | 0.099   | 0.107   | 0.117   | 0.126   |
> | R=8              | **1D-Partitioned SSDU**         | 0.083   | 0.114   | 0.134   | 0.148   |
>
> (cont' in the next response)

---

> ### Author Response · Authors · 2024-11-20
> **Response cont'**
>
> Below we report LPIPS and DISTS averaged across 100 validation samples:
>
> | **Training Data** | **Reconstruction Method**  | **LPIPS (R=2)** | **DISTS (R=2)** | **LPIPS (R=4)** | **DISTS (R=4)** | **LPIPS (R=6)** | **DISTS (R=6)** | **LPIPS (R=8)** | **DISTS (R=8)** |
> |-------------------|----------------------------|------------------|------------------|------------------|------------------|------------------|------------------|------------------|------------------|
> | R=1              | FS-DPS                     | 0.061           | 0.028           | 0.106           | 0.059           | 0.166           | 0.094           | 0.231           | 0.126           |
> | R=2              | A-DPS                      | 0.118           | 0.062           | 0.127           | 0.068           | 0.138           | 0.076           | 0.150           | 0.082           |
> | R=2              | **1D-Partitioned SSDU**    | 0.141           | 0.084           | 0.144           | 0.081           | 0.187           | 0.108           | 0.223           | 0.129           |
> | R=4              | A-DPS                      | 0.155           | 0.090           | 0.155           | 0.092           | 0.161           | 0.095           | 0.168           | 0.099           |
> | R=4              | **1D-Partitioned SSDU**    | 0.169           | 0.105           | 0.169           | 0.092           | 0.187           | 0.104           | 0.214           | 0.123           |
> | R=6              | A-DPS                      | 0.180           | 0.110           | 0.178           | 0.110           | 0.181           | 0.111           | 0.186           | 0.114           |
> | R=6              | **1D-Partitioned SSDU**    | 0.143           | 0.092           | 0.169           | 0.102           | 0.196           | 0.117           | 0.222           | 0.133           |
> | R=8              | A-DPS                      | 0.212           | 0.129           | 0.204           | 0.127           | 0.204           | 0.127           | 0.204           | 0.127           |
> | R=8              | **1D-Partitioned SSDU**    | 0.142           | 0.094           | 0.176           | 0.109           | 0.206           | 0.127           | 0.231           | 0.140           |
>
> ---
>
> **Feedback**: The paper needs to be further polished, for example, Eq 2.9 seems incorrect.
>
> **Response**: We are grateful to the Reviewer for catching this typo. We updated Equation 2.9 and Equation A.3. in the proof with the correct notation.

---

> > ### Author Response · Authors · 2024-11-24
> > **Follow-up**
> >
> > Dear Reviewer, as the discussion period approaches to an end, we would like to follow-up and check if our rebuttal adequately addressed your questions.

---

> > ### Comment · Reviewer_323G · 2024-11-25
> >
> > Thank you for revising the manuscript and incorporating an additional baseline. The experiments and comparisons with the new baseline are convincing, and I am raising my score to 6.
> >
> > I have one additional comment: the tables are not well-structured and are somewhat difficult to follow. I recommend highlighting the best performance in bold and improving the overall clarity and organization of the tables.

---

> > > ### Author Response · Authors · 2024-11-29
> > > **Response to reviewer 323G**
> > >
> > > Thank you for carefully reading our responses and for increasing the score. We have made the following changes to the formatting of all tables in the revised manuscript:
> > > 1) We added additional metrics like SSIM and PSNR to Table 2
> > > 2) We added Table 3 to report LPIPS and DISTS
> > > 3) We reorganized Table 1 for clear presentation of FID scores
> > > 4) We improved the overall structure of tables and highlighted the best performance across varying acceleration factors in bold.
> > >
> > > We hope that this is helpful.

---

> > > > ### Comment · Reviewer_323G · 2024-11-29
> > > >
> > > > Thank you for your update. I have increased the presentation score from 2 to 4.

---

> > > > > ### Author Response · Authors · 2024-11-29
> > > > > **Response to Reviewer 323G**
> > > > >
> > > > > Thank you. Please let us know if there is anything we could do to make you consider increasing further your rating. In any case, thank you, as your review helped us improve our work.

---

### Meta-Review · Area_Chair_GNSi · 2024-12-22

**Metareview:**

This paper combines the recently proposed ambient diffusion with diffusion posterior sampling (DPS) to make ambient diffusion posterior sampling (A-DPS). They adapt ambient diffusion to allow training on corrupted (say subsampled) Fourier measurements. This results in a self-supervised learning scheme that uses diffusion model. The authors apply it to MRI at different acceleration factors and find that often at high acceleration factors it is more favorable to train the models on subsampled than on clean data. The results are overall strong. Acknowledging experiments on compressed sensing, the writing and the experiments are really geared towards MRI which could have been featured in the title.

I think that this is a nice paper, but that the novelty is weak since the method combines ambient diffusion and DPS in a straightforward way and applies them to MRI. Extending ambient diffusion to Fourier is nice but technically straightforward. I agree with the authors that the finding about training on subsampled data is nice, but it alone does not immediately warrant acceptance. Given the clear writing, clear delineation of contributions, the fact that the ideas and the results are good, the relative consensus between the reviewers about the merits of this work, and the fact that it's the first work that does self-supervised diffusion for inverse problems, my recommendation is ultimately to accept.

**Additional Comments On Reviewer Discussion:**

323G thought that it's an interesting paper with strong results but criticized novelty (AD + DPS) and baselines. The authors included additional baselines and the reviewer raised score to weak accept. Authors acknowledged that AD + DPS is not particularly novel but argued that findings are interesting. GxHM praised presentation and the fact that it's a first self-supervised diffusion for inverse problems paper, but criticized novelty. They also mentioned a need for better baselines. The reviewer remained reserved regarding novelty, as well as placement if the main contribution is findings about MRI (all solid arguments!). GFMn initially had mixed feelings and criticized score estimation and the specifics of MRI experiments. After several rounds of back and forth they recommended acceptance. gieB pointed out some technical issues and asked for various clarifications. Responses pleased gieB who raised their score to 8. Q5u6 praised impact, prose, and performance, but criticized baseline end metrics. The authors gave nice responses and addressed some of the concerns. My recommendation is based on carefully studying the discussions and weighing impact (self-supervised training of diffusion models for inverse problems!) with the fact that the combination is straightforward. Impact prevailed.

---

### Decision · Program_Chairs · 2025-01-22

Accept (Poster)